# A Rate Theory Model of Radiation-Induced Swelling in an Austenitic Stainless Steel

**Malcolm Griffiths** [1,2,3,*], **Juan Ramos-Nervi** [4,5] **and Larry Greenwood** [6]

1   Department of Mechanical & Materials Engineering, Queens University, Kingston, ON K7L 3N6, Canada
2   Department of Mechanical & Aerospace Engineering, Carleton University, Ottawa, ON K1S 5B6, Canada
3   ANT International, 448 50 Tollered, Sweden
4   Departamento de Materiales, Nucleoelectrica Argentina S.A., Francisco N. Laprida 3163,
    Villa Martelli B1603AAA, Argentina; jnervi@na-sa.com.arl
5   Centro Tecnologico Aeroespacial, Departamento de Aeronautica, Facultad de Ingenieria,
    Universidad Nacional de La Plata, Avda. 1 esq. 47, La Plata B1900TAG, Argentina
6   Pacific Northwest National Laboratory, Richland, WA 99352, USA; larry.greenwood@pnnl.gov
*   Correspondence: malcolm.griffiths@queensu.ca

**Abstract:** Many rate theory models of cavity (void) swelling have been published over the past 50 years, all having the same, or similar, structures. A rigorous validation of the models has not been possible because of the dearth of information concerning the microstructures that correspond with the swelling data. Whereas the lack of microstructure information is still an issue for historical swelling data, in the past 10–20 years data have been published on the evolution of the microstructure (point defect yields from collision cascades, cavity number densities, and dislocation densities/yield strengths) allowing certain gaps in information to be filled when considering historic swelling data. With reasonable estimates of key microstructure parameters, a standard rate theory model can be applied, and the model parameter space explored, in connection with historical swelling data. By using published data on: (i) yield strength as a function of dose and temperature (to establish an empirical expression for dislocation density evolution); (ii) cavity number densities as a function of temperature; and (iii) freely migrating defect (FMD) production as a function of primary knock-on atom (PKA) spectrum, the necessary parameter and microstructure inputs that were previously unknown can be used in model development. This paper describes a rate-theory model for void swelling of 316 stainless steel irradiated in the EBR-2 reactor as a function of irradiation temperature and neutron dose.

**Keywords:** austenitic stainless steel; neutrons; radiation damage; point defects; swelling; microstructure; rate theory; modelling

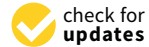



## 1. Introduction

A material issue for stainless steel (SS) components in nuclear reactors operating at temperatures ≥400 °C is radiation-induced swelling [1,2]. The effect of swelling on power reactor components operating at lower temperatures of 250–350 °C has been identified as a potential issue but the magnitude is small; swelling of baffle bolts is insignificant (<0.25%) for dose levels up to about 20 displacements per atom (DPA) and temperatures of up to about 345 °C [2,3]. The operating temperature may be as high as 420 °C at the re-entrant corners of the core baffle in pressurised water reactors (PWRs) because of nuclear heating [3] and so swelling may be a concern at those locations. Swelling is an issue for existing fast reactors operating at high temperatures and neutron dose rates and is also a concern for many new reactor designs [4]. As much as possible it is desirable to obtain a mechanistic model for swelling that can be applied outside of the range of the data from which the model was derived. Because of the difficulties in obtaining accurate swelling measurements for irradiated materials when the swelling magnitude is small (requiring

large volumes of material for accuracy), data obtained at high doses and temperatures offer the best source of accurate swelling data from which to derive a swelling model. Such data are largely derived from fast reactor irradiations.

Swelling in nuclear reactor materials is the volume expansion related to the formation of cavities (often simply referred to as voids). The volume expansion is indirectly related to cavity formation and growth. The cavities themselves do not cause a volume increase. They are simply spaces within the material where atoms have been removed. This is achieved by a slow diffusional process involving the creation of Frenkel pairs by irradiation and the subsequent migration of the vacancy and interstitial point defects. The swelling is a direct result of an excess of interstitial point defects migrating to sinks other than cavities, thus producing a volume increase. One can model swelling by considering how the flow of interstitial point defects to dislocations (which are treated as biased sinks) is affected by the microstructure (which includes grain boundary, cavity, and dislocation sink strengths), as well as temperature, neutron dose and dose rate. Cavities are treated as being unbiased sinks for interstitial point defects and are thus net sinks for vacancy point defects. Voids (cavities that are devoid of any gas) are considered neutral sinks as far as point defect diffusion to these sinks is concerned, i.e., the absorption of vacancies and interstitials is diffusion-controlled rather than reaction-rate-controlled. In the presence of gaseous impurities the resulting internal pressure imposes an extra interaction that primarily affects the net gain or loss of vacancies and this effect has to be included in any rate-theory calculation.

Cavities often contain an insoluble gas such as helium that imparts a pressure, thus preventing collapse to a vacancy dislocation loop. Cavities that do not contain an insoluble gas are called voids. Voids that are devoid of any He can be created when there is a sufficiently high vacancy super-saturation. The energetics of cluster formation often favour collapse of a void to a vacancy dislocation loop (also dependent on the loop stacking fault energy if the loop is faulted) unless the void has a low surface energy and is sufficiently large to favour void growth by the absorption of an excess of vacancy point defects [1,5–15]. According to the definition of Bhattacharya and Zinkle [1] a cavity that is empty or filled with gas below equilibrium pressure is called a void, while a pressurized cavity (near or above thermal equilibrium conditions) is a bubble. An equilibrium bubble is one where, in the absence of irradiation at a given temperature and bubble size, the work done by the pressure that would tend to cause a cavity to absorb a vacancy is balanced by the increase in surface energy that would result. During irradiation, void growth due to an excess flow of vacancy point defects is called bias-driven void growth [13]. Insoluble gases such as He stabilise cavities and can enable small cavities to grow to sizes where bias-driven growth is possible. In the absence of irradiation, at the irradiation temperature, a void will shrink until it reaches thermal equilibrium when, by definition, it is then a bubble. Both voids and bubbles are classed as cavities, which is the appropriate term to use when one does not know the gas content.

The stability of voids in the absence of helium is clearly demonstrated by observations of void swelling in electron- and ion-irradiated materials [1,16]. The nucleation of cavities is complex and dependent on the creation of a stable nucleus that is driven by either a stochastic process or the accumulation of gaseous impurities [10,11]. Stoller and Odette have modelled the growth of small meta-stable vacancy clusters as they evolve to a stable state by considering the evolution of the cluster size distribution where the clusters of a given size either grow or shrink at the expense of clusters that are one (or more) point defects larger or smaller [10–12]. To avoid the complexity of computing many coupled differential equations Stoller and Odette [11] modelled the nucleation process by assuming that the absorption of point defects at a given cluster is independent of the cluster size. They found that the nucleation time (for a given displacement damage rate) largely increased with increasing temperature for temperatures >400 °C. In later work, the evolution of the clusters was modelled without the same simplifying assumption and the nucleation rate was shown to decrease with increasing temperature for temperatures between 200

°C and 300 °C [14]. Having obtained a model for cavity nucleation rate as a function of temperature and dose rate [11], Stoller and Odette then derived a rate theory model for swelling in austenitic stainless steel as a function of temperature that provided an upper bound to the swelling data band at high fluences between 425 °C and 575 °C [12]. In modelling the microstructure evolution (dislocation loops and cavities) they made the necessary assumptions about various material parameters to obtain a result that served as an upper bound for swelling data at high fluences but gave results that tended to be shifted to higher temperatures compared to the data at low and intermediate fluences. The assumptions in choice of parameters were best estimates and the model provided partial agreement with the data.

To better understand the role of He in stabilizing cavities during irradiation it is instructive to consider why materials such as Zr are resistant to cavity formation. In contrast to stainless steels containing Fe, Cr and Ni, which are sources for He in addition to that arising from boron impurities [17–19], Zr-alloys have a very low (n,α) cross-section such that insoluble He is not readily generated during irradiation [20]. For this reason, the behaviour of Zr-alloys serves as a useful comparison to understand the behaviour of He-generating materials. Neutron-irradiated Zr-alloys tend to have vacancy loops as the most common form of vacancy cluster [21,22]. Cavity formation has been shown to occur in Zr-alloys following electron irradiation if they were also implanted with He ions, either before or during the electron irradiation [23]. Cavity formation at very high doses, ~80 DPA, without any He implantation has been shown to occur in layers parallel to the basal plane in electron-irradiated Zr at 400 °C so having He present is not necessarily a pre-requisite for swelling. In that case there were two types of cavities: (i) one set that were thin, platelet-like, and parallel with the basal plane; and (ii) another set that were small and spherical but isolated close to the thin foil surfaces that were also high in dissolved oxygen [24].

Stainless steels and Ni-alloys also exhibit swelling during self-ion irradiation (as opposed to ion irradiation with insoluble inert gas ions such as He). The fact that the swelling following self-ion irradiation of stainless steels can be large indicates that He is not necessary for swelling to occur, although there is strong evidence that helium enhances swelling [1,13,16,25].

With respect to the role of He within cavities during irradiation, cavities can be under- or over-pressurised relative to the thermal equilibrium state depending on the DPA rate, the He generation rate and the temperature [5–15]. If the irradiation is terminated, and the material is maintained at the irradiation temperature, under-pressurised cavities should shrink while over-pressurised cavities should grow. If the temperature is changed after irradiation, a new equilibrium condition is established, and the pressurised cavity will either shrink or grow according to the new pressure and temperature. Cavities that do not contain any insoluble gas (voids), on the other hand, will always shrink by emitting vacancies at any temperature. Helium is generated in all nuclear reactors so one can never be sure how much He is captured within cavities and whether the cavities are over- or under-pressurised relative to the thermal equilibrium state. Whereas Bhattacharya and Zinkle [1] define an empty cavity, or one containing gas below equilibrium pressure at the irradiation temperature, as a void, one sometimes does not know the state with respect to the equilibrium pressure. The convention adopted in this paper is therefore to refer to all three-dimensional vacancy clusters as cavities.

At very high irradiation temperatures, depending on the atomic displacement damage rate, the thermally induced vacancy point defect concentration can be so high that it dominates over any irradiation-induced point defect production. At low irradiation temperatures, vacancies can be relatively immobile to the extent that mutual recombination with self-interstitial atoms dominates the microstructure evolution of the material. At intermediate irradiation temperatures, the microstructure evolves through the diffusion of interstitial and vacancy point defects to sinks and/or clustering of these point defects creating new defect sinks. The material is then said to be in a sink-dominated condition.

The temperature at which there is a transition to thermal-dominated (higher temperature) or recombination-dominated (lower temperature) behaviour is dependent on the material (primarily the vacancy formation and migration energies), the microstructure and the atomic displacement damage rate [1,5,13,25–27]. At intermediate temperatures, in the sink-dominated regime, the properties of the material evolve according to the radiation damage in the form of displaced atoms and vacancies. Point defect clustering, in the form of dislocation loops or voids, results in a change in material properties, primarily yield strength and ultimate tensile strength [2,28] and material dimensions because of irradiation creep and void swelling [1,2]. Void swelling in austenitic stainless steels is significant at temperatures between 400 °C and 650 °C [1,2,4,5].

Swelling and embrittlement are also observed in irradiated ferritic steels [1,29,30] at roughly the same irradiation temperature ranges as for the austenitic steels. Swelling is an order of magnitude smaller in ferritic compared with austenitic steels at similar doses. The lower swelling has been related to strong interactions between solute atoms and point defects in ferritic steels, but microstructure differences may also play a role. Radiation effects are often very sensitive to vacancy diffusion, vacancy mobility being a limiting factor in mass transport. Vacancy migration energies used in modelling are 1.28 eV in ferritic steels [30] and 1.38–1.4 eV in austenitic steels [31,32].

The principles of swelling have been established over many years and there are many papers describing rate theory models of irradiated materials. However, many of the models that have been developed are not specific to a particular set of data and are mostly generalised, addressing the dependencies of swelling on various parameters and comparing with select sets of data. The aim of this paper is to use a specific set of good quality swelling data as a function of temperature for the same material and irradiated in the same reactor to develop a rate theory model and define key generic material parameters that can then be applied in predicting swelling for other materials irradiated in different reactors.

## 2. Materials and Irradiation

To develop a rate-theory model that can then be applied to different materials at different temperatures and irradiation conditions, the structure of the model and key parameters must be validated using well characterised data. Swelling data for 20% cold-worked 316 stainless steel (SS) irradiated in experimental breeder reactor-2 (EBR-2) will be used for this purpose. These data have been reported previously and describe the evolution of swelling with fast neutron dose at temperatures between 400 °C and 650 °C [2]. Although the doses for each measurement have been reported as fast neutron fluence (E > 0.1 MeV) there were no specific details concerning neutron dose rate and He generation rate. Knowing that the material was 316 SS irradiated in Row 2 of EBR-2 and that the samples were subject to a range of neutron dose rates that were constant at any one temperature [33], it is then possible to determine the atomic displacement damage rate and He generation rate for each temperature set using the spectrum applicable to Row 2 at the peak (reactor mid-plane) position. With additional data on microstructure evolution (dislocation and cavity microstructure) and advances in the calculations of freely-migrating defect (FMD) production rates, there is sufficient information available to develop a rate theory model using this one set of data that can then be applied to other material irradiations. The swelling data for 20% cold-worked 316 SS irradiated in EBR-2, Row 2 published by Garner and Gelles [33] is shown in Table 1.

**Table 1.** Dose and swelling data for 20% cold-worked stainless steel irradiated in Row 2 of EBR-2. Fluences are given as E > 0.1 MeV. Dose rates vary for each temperature.

| 400 °C | | 482 °C | | 538 °C | |
|---|---|---|---|---|---|
| Fluence | % Swelling | Fluence | % Swelling | Fluence | % Swelling |
| 6.76 | 0.12 | 4.25 | 0.16 | 5.56 | 1.45 |
| 14.07 | 1.54 | 9.16 | 5.97 | 9.27 | 7.71 |
| 15.71 | 2.61 | 12.11 | 10.29 | 13.75 | 15.28 |
| 17.89 | 4.54 | 15.27 | 21.79 | 17.02 | 27.65 |
| | | 19.85 | 45.92 | 23.02 | 56.56 |
| **427 °C** | | 21.93 | 55.27 | 26.29 | 72.21 |
| Fluence | % Swelling | | | | |
| 3.82 | 0.16 | | | | |
| 5.13 | 0.14 | **510 °C** | | **593 °C** | |
| 8.51 | 1.18 | Fluence | % Swelling | Fluence | % Swelling |
| 12.11 | 3.09 | 5.67 | 0.57 | 6.11 | 1.66 |
| 16.04 | 6.74 | 11.13 | 10.96 | 10.15 | 8.14 |
| 20.51 | 18.67 | 14.07 | 22.68 | 14.84 | 16.13 |
| 23.35 | 26.25 | 17.56 | 41.16 | 18.00 | 24.15 |
| | | 23.78 | 70.50 | 25.09 | 49.34 |
| **454 °C** | | 27.16 | 86.36 | 28.58 | 64.11 |
| Fluence | % Swelling | | | | |
| 9.82 | 0.73 | | | **650 °C** | |
| 12.87 | 7.01 | | | Fluence | % Swelling |
| 15.16 | 20.05 | | | 6.00 | 0.35 |
| 17.45 | 30.26 | | | 14.73 | 3.93 |
| | | | | 17.78 | 6.72 |
| | | | | 24.65 | 18.61 |
| | | | | 27.93 | 26.41 |

## 3. Methodology and Analysis

### 3.1. Atomic Displacement Damage and He Gas Production

At a given temperature, irradiation-induced swelling is primarily a function of the number of atomic displacements. The irradiation damage dose is represented by displacements per atom or DPA, which is a measure of how many times every atom is displaced from its lattice site. For PWR core components made of austenitic stainless steel every atom is displaced about two times every year (2 DPA/year) when averaged over the core volume. A fractional measure of dose, for example 0.1 DPA, means that one in every 10 atoms will have been displaced over the period of the dose measure.

Although most empirical data on swelling are reported in terms of fast neutron fluence (E > 0.1 MeV or E > 1 MeV), some data are reported in DPA that accounts for the spectral differences between different reactors. When DPA values are quoted they refer to the primary atomic displacements within collision cascades [5]. When modelling swelling, the radiation damage input to the calculations is in the form of FMDs. These are the mobile clusters and mobile point defects that survive the collision cascade and can migrate to sinks such as dislocations or cavities. Whereas point defects are known to be mobile and one can calculate their migration to different sinks, little is known about the properties of mobile clusters and so most calculations simply assume that all mobile radiation damage is in the form of point defects, i.e., self-interstitial atoms and vacancies. Models based on point defect diffusion, including that of gaseous atoms, which assist with stabilisation of large three-dimensional vacancy clusters (cavities), thus depend on knowing the gas production rate (primarily He) and the percentage of displaced atoms and associated vacancies that are free to migrate after the collapse of the atomic displacement (collision) cascades, the so-called freely-migrating-(point)-defects or FMDs.

Apart from the effect of atomic displacements caused by energetic neutrons, irradiation can affect material properties by transmutation that creates a new isotope or element [17,20]. When the elements in the alloy have a high neutron capture cross-section for either (n,γ), (n,α) or (n,p) reactions a large number of additional displacements can be produced. Most

of the transmutation-induced displacements are caused by the energetic recoil when the particle or photon is emitted but there are some additional displacements that can arise from the emitted particle or photon [34–37]. Transmutation generally does not have a large effect on material chemistry in engineering alloys such as austenitic stainless steels, either because the transmutation is to another isotope of the same element or the chemical changes induced by transmutation reactions are small [20].

Some transmutation processes, i.e., those involving the production of He or H via $(n,\alpha)$ or $(n,p)$ reactions, can have a large effect on material properties because of the emitted particle itself, the most notable being the effect of He resulting in enhanced swelling and a loss of ductility known as He-embrittlement [25,38–50]. Neutron-irradiation-induced He production comes from $\alpha$-particles, which are ionised He atoms. He-embrittlement is typically observed at temperatures that are overlapping but slightly above the main swelling regime, i.e., between 500 °C–700 °C [2,40–47]. The loss of ductility increases with temperature and He concentration, represented by [He], [47]. The embrittlement has been largely attributed to segregation of He-stabilised cavities at grain boundaries; the extent of cavity coverage increasing with both temperature and He concentration. Whereas cavities in the matrix will harden the material, the other main effect of cavity formation and growth is on the dimensional stability of the material, i.e., swelling.

In an earlier attempt at modelling the swelling data reported by Garner and Gelles [33] the irradiation conditions were unknown and the rate theory calculations were performed with an assumed spectrum, that of EBR-2 row 8, location D5 [51]. It is now known that the irradiations were performed in row 2, which has a different spectrum and a higher atomic displacement and gas production rate. Because of potential spectral and dose-rate effects new calculations with the appropriate spectrum provide more accurate predictions and eliminate one source of uncertainty. Using a representative spectrum for EBR-2, row 2 at the reactor midplane the atomic displacement damage and gas production have been calculated using the SPECTER code [34]. By applying the methodology outlined by Greenwood and Garner for determining the transmutation effects for DPA, [H] and [He] as a function of operating time [35,36] both the atomic displacement rate and gas production rates have been calculated. The two spectra are shown in Figure 1 for comparison.

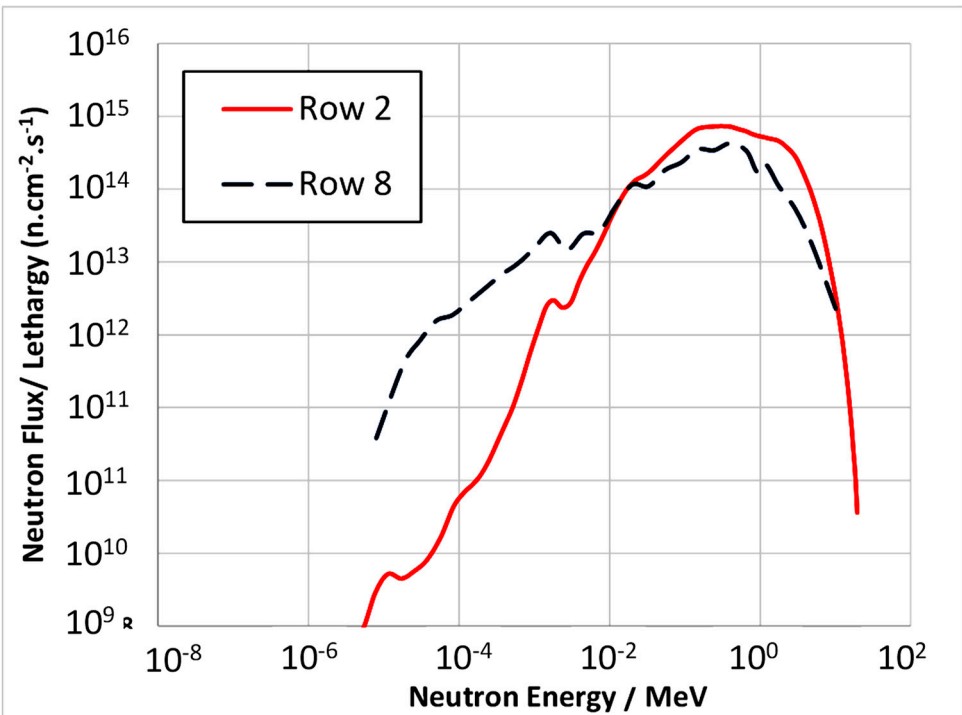

**Figure 1.** Neutron spectrum represented as flux per unit lethargy decrement in EBR-2 row 2 at the centreline and row 8 D5.

The nominal compositions in wt% and at% for 316 SS used for the damage calculation are shown in Table 2.

**Table 2.** Nominal composition of the 316 stainless steel.

| Material | Ni | Cr | Fe | C | Mn | Ti | Si | Mo |
|---|---|---|---|---|---|---|---|---|
| 316 (wt%) | 12 | 17.0 | 67.3 | 0.08 | 2 | 0.5 | 1.0 | 2.5 |
| 316 (at%) | 11 | 17.6 | 64.5 | 0.4 | 2 | 0.6 | 1.9 | 1.4 |

Using the compositions as shown, the atomic displacement damage, He and H production for 316 SS for row 2 (midplane) and row 8 (D5) in EBR-2 as a function of operating time are shown in Figure 2. Assuming the peak fluence reported for the AA-1 experiment [33] for an irradiation temperature of 593 °C corresponds with row 2 at the midplane then the corresponding fluence data for all other temperatures (each irradiated at a constant neutron flux for the same time) provides a scaling factor from which to deduce the corresponding radiation damage rates that are required inputs to the model. Any variations in damage rate can then be accurately accounted for.

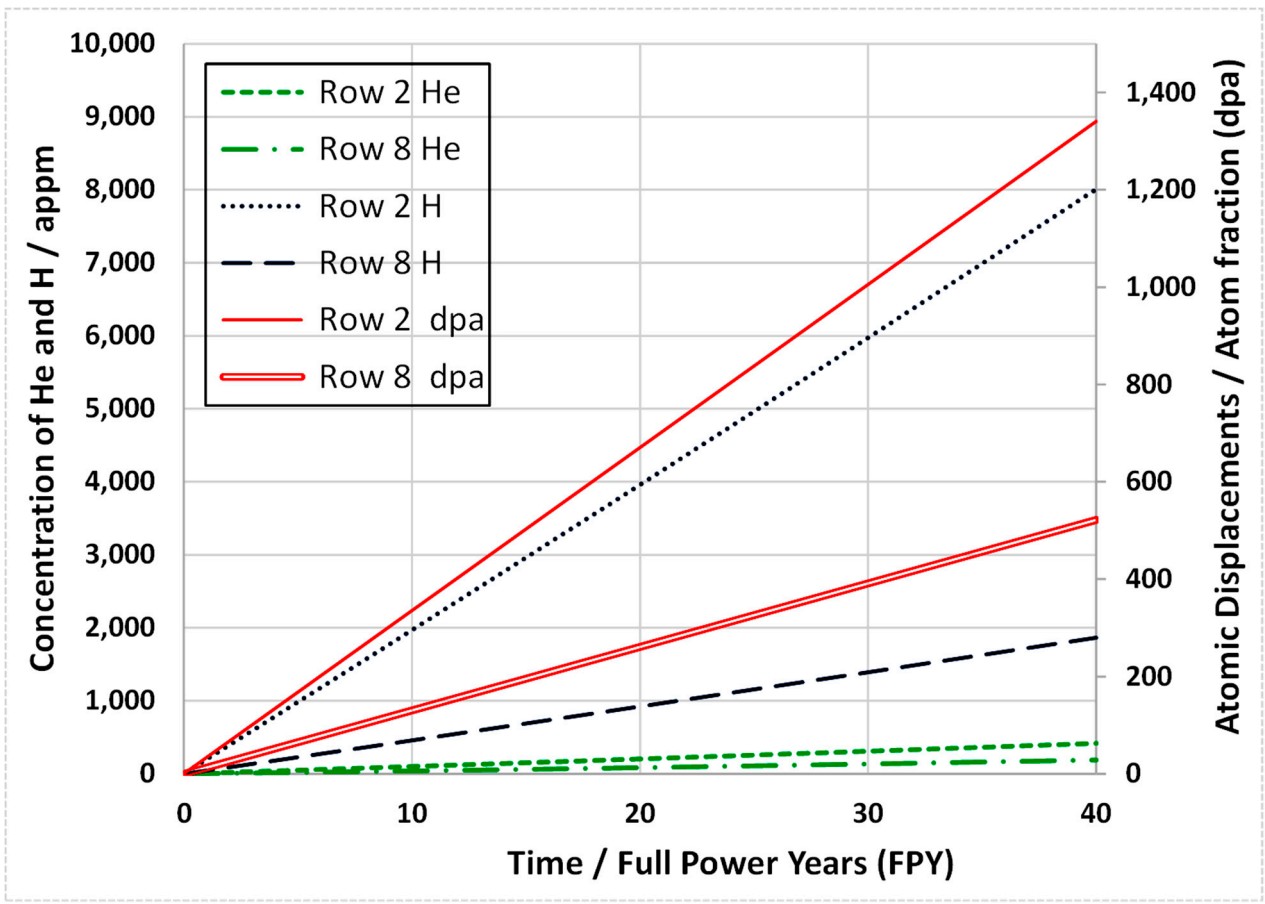

**Figure 2.** Atomic displacement damage (DPA), [He] and [H] as a function of operating time for 315 SS in EBR-2 row 2 (midplane) and row 8 (D5). The He/DPA ratios averaged over 40 years are 0.30 and 0.29 for row 2 and row 8, respectively.

Although the hydrogen generation is considerably higher than that of He, and there are some data to indicate that some of the hydrogen may be retained in the material rather than released into the environment [52], there are no data to show that hydrogen has an effect on swelling (or other properties) at the temperatures being considered here (400 °C–650 °C). Hydrogen is generally considered to have a direct effect on material properties in austenitic alloys at temperatures <150 °C [53]. As hydrogen and helium

generation are produced coincidentally, any effect of hydrogen on swelling is hard to separate from that of helium. The role of He in swelling is well known but there are no definitive experiments where the role of atomic hydrogen, produced during irradiation, on swelling is clearly determined. In the absence of any definitive data, hydrogen will therefore not be included in the modelling of swelling presented here.

*3.2. Freely Migrating Point Defects*

Collision cascades produce a high density of point defects that can cluster and spontaneously recombine as the displacement spike cools. Although clusters formed in the cascade can contribute to radiation-hardening, other processes such as irradiation-induced creep, swelling and micro-chemical segregation are expected to be largely dependent on the freely migrating [point] defects. The interstitial and vacancy point defects that constitute the FMDs can themselves recombine, either mutually or because they migrate to the same sinks. This additional recombination at sinks depends on the availability of both neutral and biased sinks for the point defects in the microstructure. The balance between recombination and annihilation at sinks that gives a net measurable effect (such as swelling) is dependent on the FMD production rate and the irradiation temperature. Typically, FMDs left over from the collapse of collision cascades constitute a small fraction of the total number of atoms displaced within the cascades [5]. The calculation of the FMD fraction as a function of primary-knock-on atom (PKA) energy has been performed here by starting with the data from Gao et al. [54] for the total residual displaced atoms after cascade collapse as a function of PKA energy. The Gao formula includes clusters and FMDs. Wooding et al. [55] suggested that less than 1/3 of the surviving defects are in the form of FMDs. The portion of the total surviving defects that are in the form of FMDs can be determined based on how well the data can be fit using a model for irradiation processes such as swelling, or radiation-induced/enhanced segregation, by scaling the cascade efficiency term. The FMD production relative to the calculated primary DPA, also known as the Norgett-Robinson-Torrens (NRT) DPA [56], is often referred to as the FMD fraction. The FMD fraction, as it is known, is a function of the PKA spectrum that is, in turn, a function of the neutron spectrum [34]. There are a number of studies that exist concerning the FMD production efficiency as a function of PKA energy [54–64].

In the first instance there are molecular dynamics (MD) simulations that provide the defect production that includes both mobile defects and defects trapped in clusters after the cascade collapse [54]. More recent MD modelling for a gas-cooled power reactor spectrum has provided information on the damage efficiency that also includes the FMD production as a function of temperature using experimental data on phosphorus segregation to grain boundaries for $\alpha$-Fe [58]. The latter work shows that there is a negative temperature dependence for FMD production efficiency for PKA recoil energies >1 keV, i.e., the FMD fraction is higher at lower temperatures. Faulkner et al. [58] provide an expression for this temperature dependence that is applicable to the PKA spectrum used for their simulation and is also applicable to $\alpha$-Fe that has a body-centred-cubic (BCC) crystal structure. They show that between 100 K and 900 K the FMD production efficiency drops from about 18% to 13%. Faulkner et al. note that the range of FMD fraction values that they have derived is larger than the values that other workers have used to account for experimental observations on chemical segregation during irradiation [64].

The FMD production efficiencies have also been studied experimentally for Ni-alloys by Rehn et al. [61,62] between 350 °C and 1200 °C and Okamoto et al. [63] between 500 °C and 600 °C. Their data showed very little effect of temperature on FMD production at the elevated temperatures being studied. Petrovic et al. [59] and Kwon and Motta [57] have reported a formula for FMD efficiency applied to stainless steels that is based on a combination of data from Rhen et al. [61–63] and Naundorf et al. [64]. The Naundorf data cite experimental data suggesting that the normalisation factor for cascade efficiency from Rehn et al., which is normalised to 1 MeV protons, is 0.2. The Naundorf factor of 0.2 is taken into account by Kwon and Motta [57] but the source of the 0.2 factor is not apparent

in the data from Rehn et al. [61,62] or Okamoto et al. [63]. In fact, Okamoto et al. [65] state that "by using electrons or protons for which [the cascade efficiency] $\varepsilon = 1$, the absolute efficiency of any particle can be determined." As their data, which gave FMD production efficiencies for various primary recoil (PKA) energies, were normalised to 1 MeV protons for which $\varepsilon = 1$, that normalisation factor will be used here to determine the FMDs as shown in Figure 3.

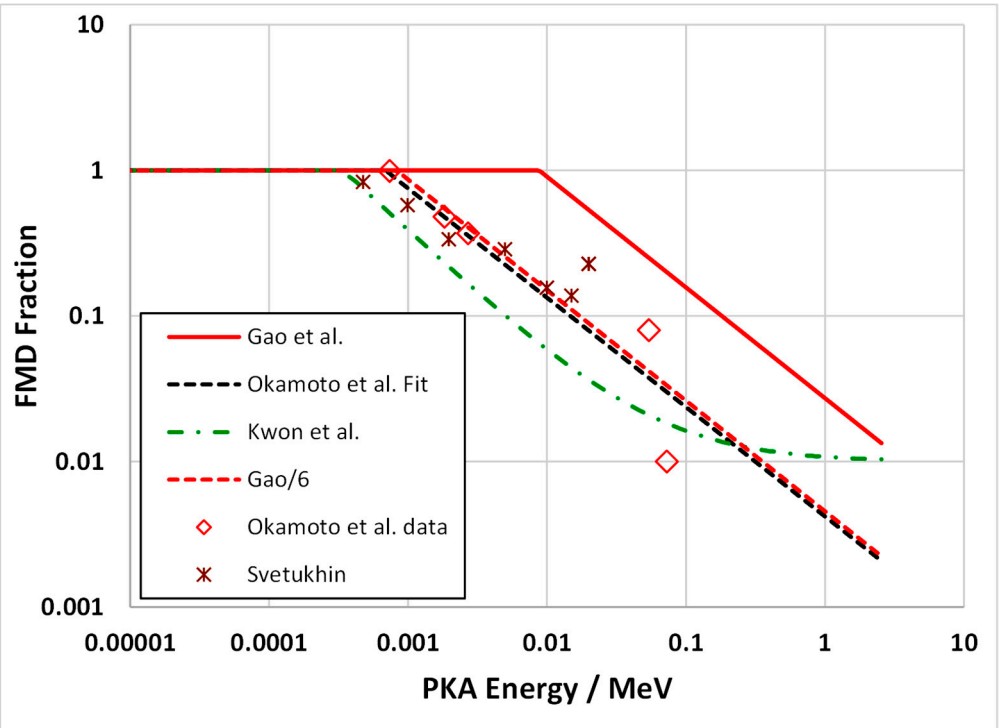

**Figure 3.** FMD fraction for Ni as a function of cascade energies for the different PKA values.

The data from Okamoto et al. [63] are compared with the formula proposed by Kwon et al. [57] and the formula proposed by Gao et al. [54] to account for all surviving defects (FMD and clustered) that would be applicable to 316 SS being irradiated between 400 °C and 650 °C in Figure 3. Additionally shown in Figure 3 are recent data from Svetukhin et al. [66] for PKA cascade efficiencies in Zr, which conform with the fit to the data of Okamoto et al. [63]. It is apparent that the functional form of the fit to the data of Okamoto et al. [63] follows the same trend as the fit from Gao et al. [54] in terms of the relative dependence on PKA energy. The difference between the experimentally derived data of Okamoto et al. [63] and the molecular dynamics simulation of Gao et al. [54] is that the former is for FMDs and the latter refers to all surviving defects whether in the form of mobile defects or immobile clusters. This agreement between the trends from the two sets of data lends support to the use of a fit to the data of Okamoto et al. [63] to provide a value for FMD fraction that is applicable to the EBR-2 data.

When calculating FMD production one has to consider the effect of gamma photons in damage production because they produce damage via a two-stage process involving the creation of energetic electrons that then displace atoms, mostly as Frenkel pairs. There is no in-cascade spontaneous recombination compared with the higher energy displacement cascades produced by high energy neutrons. The efficiency of production of FMDs for gamma photons is therefore 1, or 100%. The inclusion of gamma photons is only necessary when the atomic displacement damage rate from neutrons is comparable with that from gamma photons, i.e., in peripheral regions of reactors where the high energy neutron flux can be low while the gamma flux is high [59,60]. For fast reactors such as EBR-2 the

gamma damage is likely small relative to the damage from high energy neutrons and can be ignored when addressing swelling at high neutron doses [51].

Using the FMD fraction as a function of PKA energy derived by Okamoto et al. [63] the effect of PKA energy spectrum on the production of FMDs is illustrated for Fe atoms in EBR-2 in Figure 4. The plots show the PKA spectrum generated by the SPECTER code [34] and a damage energy parameter (TDAM), which is the value of the energy available to create atomic displacements as defined by Norgett et al., [56]. The DPA is determined by applying TDAM to each PKA. The FMD for neutrons is obtained by applying the cascade efficiency (Figure 3) to the DPA produced for each PKA energy. Similar calculations for the other main elements in 316 SS (Ni and Cr) have provided an FMD fraction value that can be applied to the primary DPA values for 316 SS in EBR-2. For atomic displacement damage that comes from the high energy PKA recoils due to He or H atom production, the FMD fraction is small (about 1%) [51]. The effect of damage from $^{59}$Ni, which can generate high energy recoils from the (n,p) and (n,$\alpha$) reactions has been shown to be a small fraction (<0.01%) of the total damage production for stainless steel in EBR-2 [51] and is ignored for the purposes of this study. For Ni-alloys in a high thermal neutron environment the mode of damage production can have a large impact on the calculated damage efficiency, which will be considerably lower than calculated from neutrons when there is a large fraction of damage coming from high energy (n,$\alpha$) reactions. Factoring in the contributions from Fe, Cr and Ni, the primary DPA rate for the midplane of row 2 in EBR-2 is $10.4 \times 10^{-7}$ s$^{-1}$. The corresponding FMD rates using the models from Kwon and Motta [57] and Okamoto et al. [63] are 5 and $9.9 \times 10^{-8}$ s$^{-1}$ giving FMD damage efficiencies of 4.8% and 9.5%, respectively.

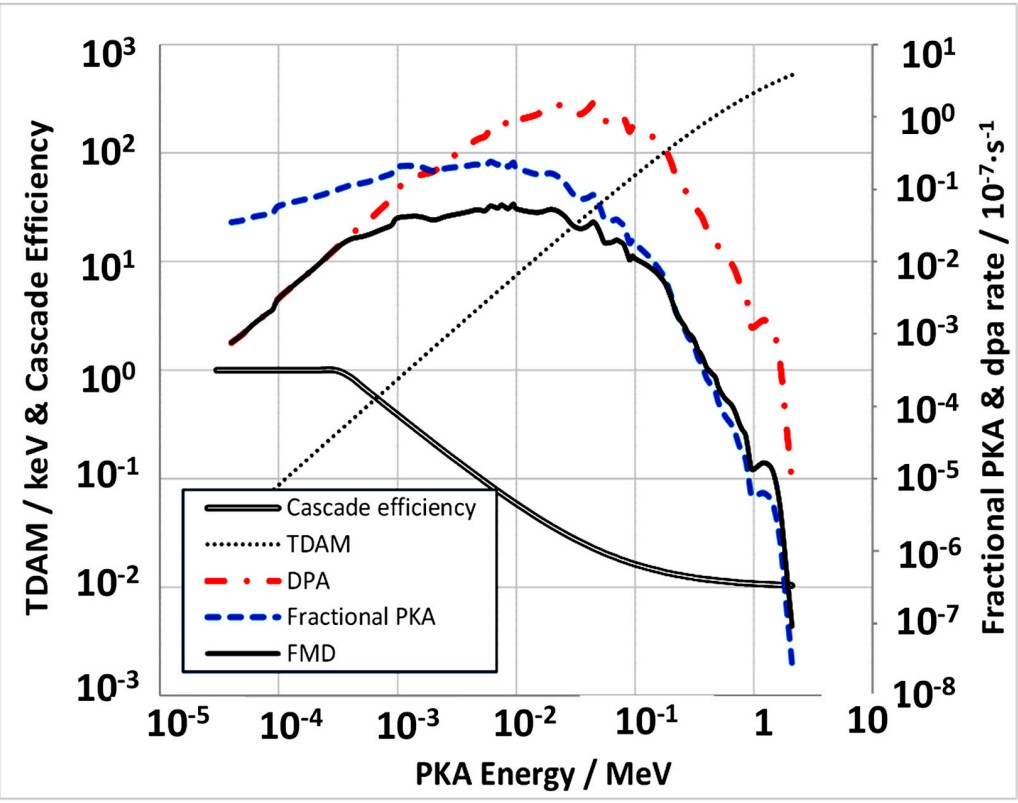

**Figure 4.** PKA spectrum, DPA and FMD for Fe in EBR-2, Row 2, reactor midplane, as a function of cascade energies for the different PKA values.

## 4. Microstructure Parameters

### 4.1. Dislocation Density

The dislocation structure in irradiated stainless steels is comprised of a mixture of network dislocations and dislocation loops formed by the clustering of interstitial point defects. The main role of the dislocations structure in swelling is through the effect of dislocations as biased sinks for interstitial point defects [67]. The existence of a biased sink for interstitial point defects allows neutral sinks such as voids to grow because of a bias-driven net absorption of vacancies [13]. Dislocation loops are not equivalent to network dislocations in terms of their elastic interactions with interstitial point defects because the stress field from opposite sides of the loops tends to cancel at large distances from the loop. Dislocations have an additional role in trapping He, which is manifested as enhanced cavity nucleation on dislocations [1]. The trapping of He at dislocations may be one reason why the onset of void swelling is delayed when a material is cold-worked [2,68]. Garner has stated that "most researchers have concentrated on the cold-work level as the primary way to delay void swelling by extending the transient regime" while recognising that recovery and recrystallisation occurs depending on the level of cold-work and the irradiation temperature [2]. Rather than using cold-work as a measure of the relative dislocation density it is best to have some measure that directly applies to the material of interest that is being irradiated. In the absence of direct measurements of dislocation density using either TEM or XRD, which have their own shortcomings, a reasonable surrogate measure that is readily available and can give some idea of the relative dislocation densities in stainless steels, for which swelling data exist, is the yield strength [2].

Data from Garnier et al. [69] for stainless steels showing the dose dependence of the yield strength at 330 °C from low doses is assumed to serve as a surrogate for dislocation density evolution where the dislocations structure evolves to a steady-state condition and remains constant at high doses [2,70]. The data can be represented mathematically with a cluster dynamics model [12]. The evolution can also be represented empirically by using a logistic-type empirical model [51]. A fit to Garnier yield stress data is shown in the supplementary materials, Figure S1. The magnitude of the dislocation densities can be scaled by the relative maximum yield stresses but the absolute magnitude is not known and can be considered as an input variable when including such data in a rate theory model.

Yield stress data for irradiated 20% cold-worked 316 SS as a function of temperature and neutron dose are shown in Figure 5. These data show that irradiation at high temperatures does not automatically result in a higher dislocation density because of point defect clustering in the form of dislocations loops. In fact, the yield stress generally decreases with increasing temperature ≥500 °C indicating that recovery dominates over any point defect clustering at these elevated temperatures. At temperatures <500 °C the elevation of the yield stress from radiation damage is inversely related to the irradiation temperature. Given that the structure is likely comprised of a mixture of line dislocations and dislocation loops, the determination of the sink strength will be confounded by the difference in long-range stress fields from the two types of dislocation. Apart from recovery of network dislocations, lower dislocation loop densities at high temperatures are presumably related to a reduction in point defect super-saturation at the higher temperatures in addition to the effect of temperature on the binding of point defects [5].

Any hardening effects resulting from the irradiation are progressively lessened as the temperature increases and at temperatures >500 °C can result in recovery of the original cold-worked dislocation structure, i.e., a reduction in the yield strength, whether or not the material is being irradiated [2] as shown in Figure 5. Extracting the yield stress data from the plot one can derive a fit to the maximum yield stresses as a function of temperature and this can be matched to a hypothetical effective dislocation density as shown in Figure 6. The magnitude has been chosen to give the best fit when applying the rate theory model given the dislocation bias parameter calculated according to Heald and Speight [67]. Even though the magnitude of the dislocation densities seems low for 20% as-cold-worked material,

one should bear in mind that samples from any engineering component are likely to have been heat-treated or stress-relieved after manufacture. The additional time in reactor at an elevated temperature will also add to the recovery of the as-cold-worked dislocation structure before one even starts to model irradiation effects.

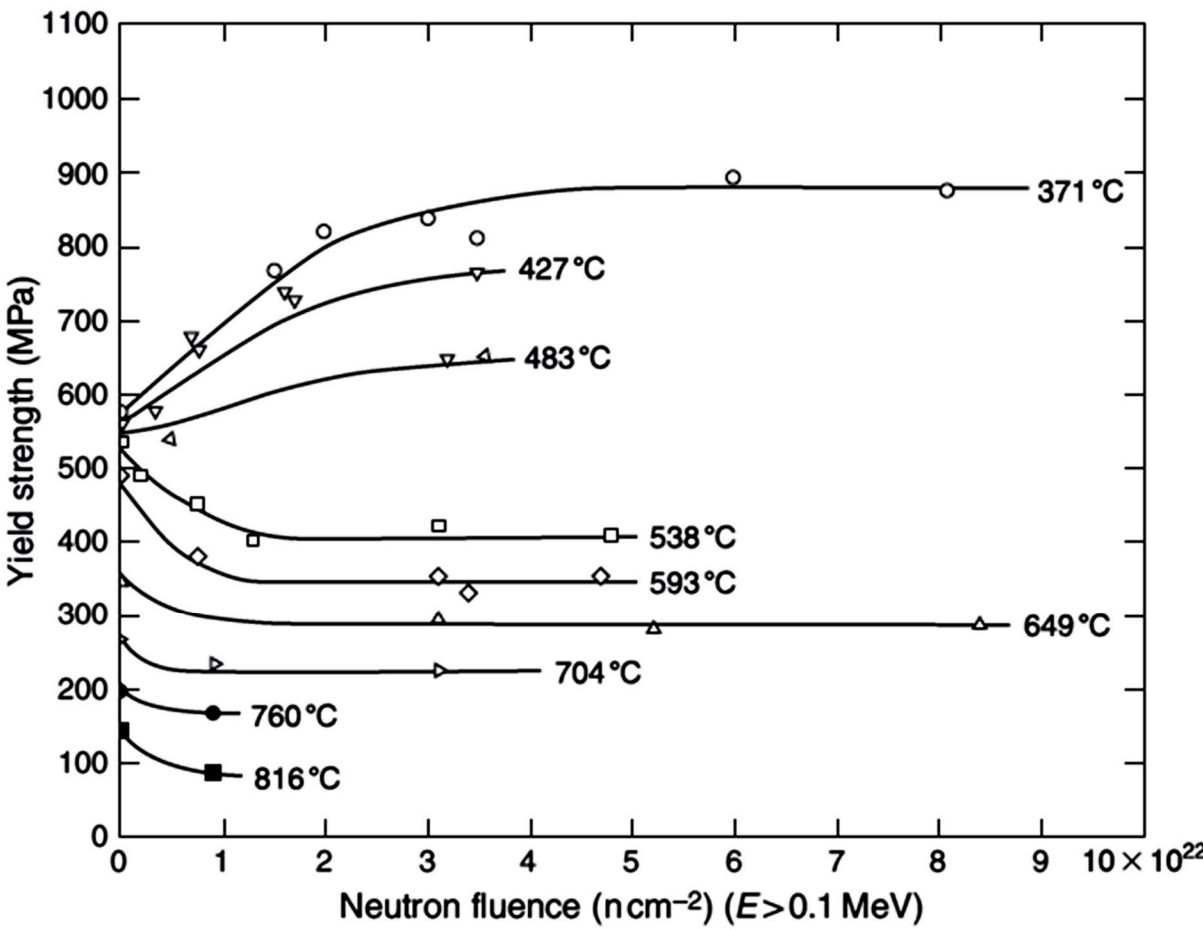

**Figure 5.** Evolution of yield strength in 20% cold-worked 316 stainless steel irradiated in EBR-2 over a wide range of temperatures. Tests conducted at the irradiation temperature. Reproduced (with permission) from [70]. Copyright Elsevier, 1981.

Interpreting the data shown in Figure 6 in terms of a dose dependent evolution using a logistic model, the dislocation sink strength (based on the dislocation density) can then easily be applied in a rate-theory model. The assumed dislocation density evolutions as a function of dose for each temperature are shown in Figure 7. The magnitude of the dislocation density is consistent with data reported by Maziasz and Busby [28] for network dislocations in 25% cold-worked primary candidate alloy (PCA), which has a similar composition to 316 SS, at different irradiation temperatures. It is assumed that the dislocation density is the same for all specimens in the as-fabricated condition at room temperature ($2 \times 10^{14}$ m$^{-2}$) and evolves as a function of temperature and dose as shown in Figure 7. The trends shown indicate how the dislocation densities change with increasing dose (time) at each temperature. The values chosen, as we shall see, give reasonable agreement with the swelling data when included in a rate theory model. The best agreement is obtained for a given cavity density (based on empirical data) and is subject to the choice of theoretically derived interstitial bias parameter [67] (see Section 5). Even if the magnitude of the effective dislocation density, i.e., that which is used to derive a sink strength, is too low, the relative effect for a given cavity structure is also dictated by the choice of dislocation bias parameter for the absorption of interstitial point defects. As there are uncertainties in both the magnitude of the dislocation density and the bias

parameter [14], one can opt to fix one and change the other depending on how well the model fits the data.

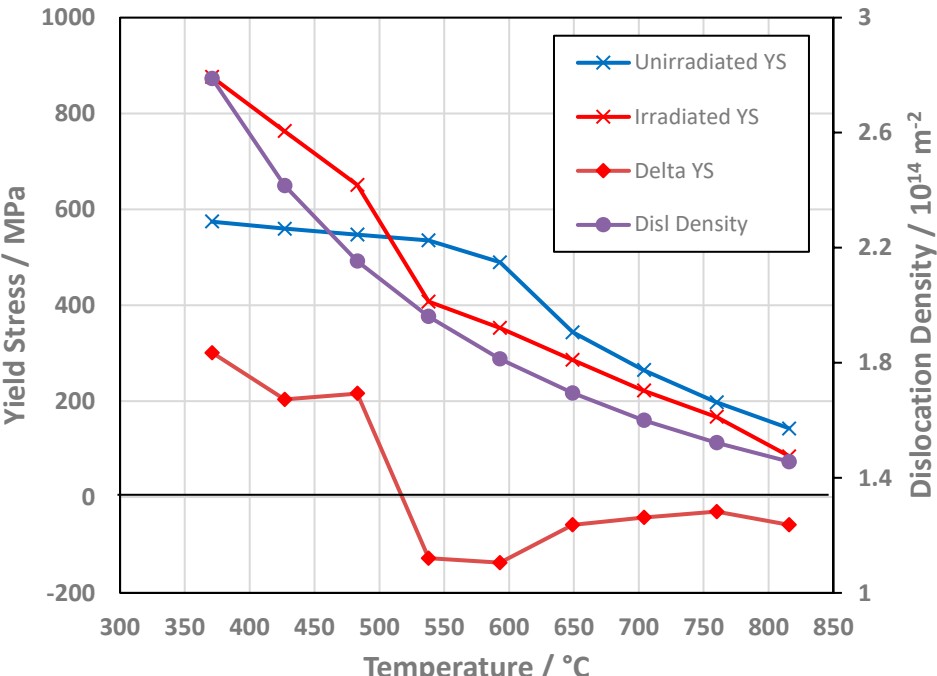

**Figure 6.** Yield stress (YS) of 20% cold-worked 316 SS determined from post-irradiation testing at the irradiation temperature after the material has reached a steady state condition in EBR-2. The assumed dislocation density of 316 SS as a function of irradiation temperature is shown on the secondary *y*-axis.

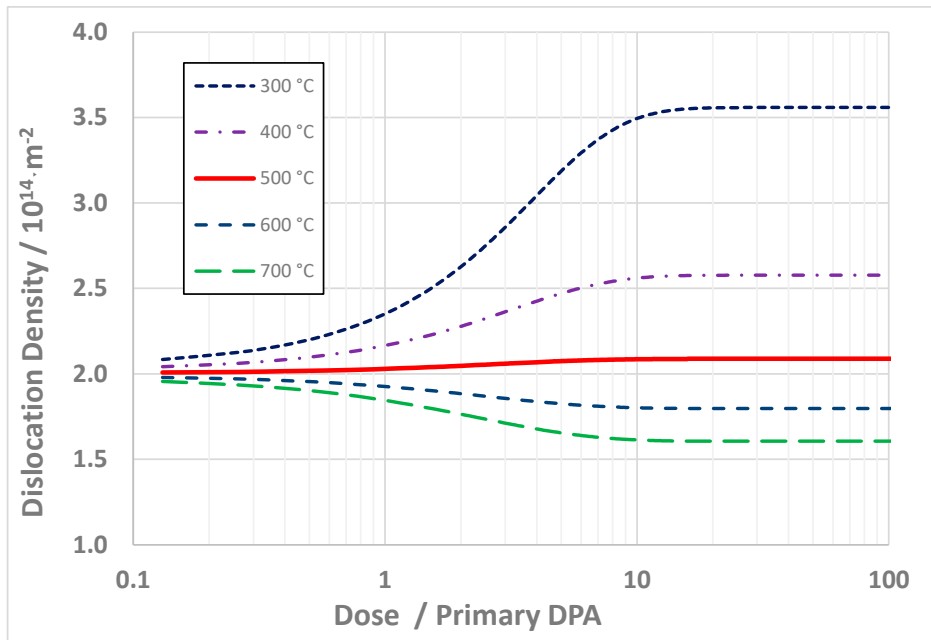

**Figure 7.** Hypothetical variation in dislocation density of 316 SS as a function of irradiation temperature during irradiation in EBR-2.

The corresponding interstitial bias calculated using the formulae developed by Heald and Speight [67] for dislocations is a function of the temperature and dislocation density. Using their formulae, the evolution of the interstitial bias as a function of dose is given in the supplementary materials, Figure S2.

### 4.2. Cavity Size and Number Density

For a given dislocation microstructure, an important parameter needed to model cavity swelling is the cavity number density, which is established in the early stages of the irradiation [14]. The uncertainties involved in modelling of the cavity nucleation [11] can be circumvented to some extent by using recently compiled data on cavity number densities from various sources as a function of temperature [1]. By assuming these are steady state number densities that represent the number densities that exist once the nucleation rate has decreased to low values at low irradiation doses (<10 DPA), which is also consistent with the modelling results [14], one can avoid the cavity nucleation problem by assuming that the number density measured at high doses is the same as the number density of cavity nuclei established at a low doses. In the model presented here for fast reactor irradiation, nucleation is thus deemed more-or-less complete at a dose of 3–4 DPA [11].

One important assumption to be used in the model is that the cavity number density is established early in the irradiation (for doses less than about 3 DPA) and thereafter remains constant. The swelling is then calculated based on the growth of the cavities. The choice of fixing the number density and allowing the cavity size to increase with dose is made for expediency and is justified because the cavity sink strength for point defects is linear with number density and cavity diameter [67] so varying one or the other by the same factor has no impact on the result (see Section 5). Moreover, data compiled by Bhattacharya and Zinkle [1] on cavity diameters and number densities in austenitic 316 SS for irradiations in fast reactors shows that the diameters are increasing with increasing dose, Figure 8. The swelling is therefore assumed to be dependent on the growth of the cavities, which increase in diameter as a function of dose, for a constant number density at a given temperature. As we shall see in Section 6, the swelling data reported by Garner and Gelles can be bounded with a model by using the two sets of cavity densities indicated by the dashed lines in Figure 8b. The two red-dashed lines represent the condition that is either: (i) mid-way between cavities and bubbles, i.e., with densities that are a factor of 10 larger than the fitted line for voids or (ii) in the upper part of the void data with densities that are a factor of 5 larger than the fitted line for voids. As we shall see later, one can then use the cavity size evolution shown in Figure 8a as a means of verification of the model output.

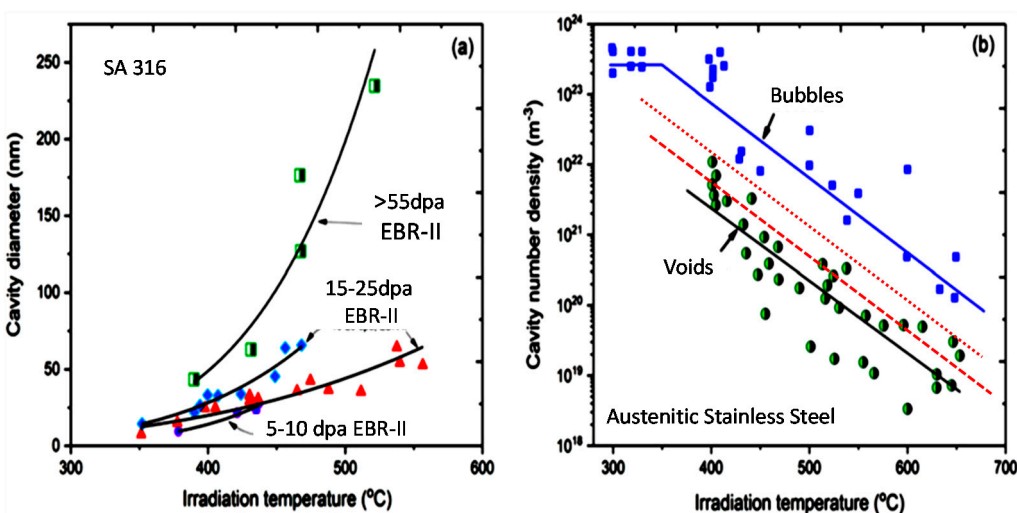

**Figure 8.** (**a**,**b**) Influence of neutron irradiation temperature on cavity diameter and number density in 300 series austenitic steels. Reproduced (with permission) from [1]. Copyright Elsevier, 2020.

The line fitted to the void densities (under-pressurised cavities) in Figure 8b are consistent with other data from EBR-2 for Ni-Fe-Cr ternary alloys having a low Ni content (19 wt%) reported by Garner [2] and Hamilton et al. [48]. There are two separate sets of data in Figure 8b, one for voids and one for bubbles. Even though EBR-2 is a liquid metal fast reactor the value for He/DPA is not negligible. Using the spectra and calculated He

and DPA values shown in Figures 1 and 2 for 316 SS, the He/DPA values are 0.3 and 0.29 averaged over 40 years for rows 2 and 8 in EBR-2, respectively. It is worth noting that for pure Ni and Fe the He/DPA values are 1.68 and 0.11, respectively, in row 2. The high value for Ni is because of the relatively high (n,$\alpha$) cross-sections for naturally occurring Ni-isotopes at neutron energies > 1 MeV [27]. Thus, although Ni-alloys have a propensity for He production in thermal neutron environments because of the thermal neutron absorption by $^{58}$Ni producing $^{59}$Ni that has a high (n,$\alpha$) cross-section over a large energy range [35,36], naturally occurring Ni will still produce more He than Fe or Cr in a fast reactor where there is a high flux of high energy neutrons.

## 5. Swelling Model Development

### 5.1. Swelling Data

Data on swelling in austenitic alloys such as 316 SS and Ni-alloys covering temperatures up to 650 °C have been reviewed and summarised by Garner [2]. Garner has shown that, for cold-worked 316 SS, the swelling rate at high fluence tends to 1% per DPA between 450 °C and 500 °C, which encompasses the peak swelling temperature, see Figure S3 in supplementary materials. The swelling at 650 °C is lower, approaching that of 0.2% per DPA, and close to the swelling rate at 400 °C.

There is also a relationship between swelling and Ni content, there being a minimum in swelling observed for Ni concentrations between about 30–50 at % Ni. The reason for the reduction in swelling up to 30% Ni is open to speculation and is largely not understood [2,26]. The negative effect of increasing [Ni] on swelling up to about 40 wt% Ni is observed for both self-ion irradiation and for neutron irradiation, the latter creating He in proportion to the Ni concentration [27]. The increase in swelling at high Ni concentrations for neutron irradiation is likely to be the result of the effect of increased He production due to the high value of the (n,$\alpha$) cross-section for $^{58}$Ni at high neutron energies, as well as $^{59}$Ni produced by the $^{58}$Ni(n,$\gamma$)$^{59}$Ni reaction. These elevated contributions to [He] only occur for neutron irradiations [27]. Whereas very high helium concentrations can be produced in PWRs and BWRs due to the high thermal neutron cross section for the $^{59}$Ni(n,$\alpha$) reaction, the main contribution to He in a fast reactor comes from the high neutron flux at high neutron energies and the high (n,$\alpha$) cross-section for $^{58}$Ni. In fast reactors, the helium production from $^{59}$Ni increases as the irradiation dose increases and the neutron spectrum softens. For row 8 in EBR-2 the He/DPA increases from 0.27 to 0.36 over a 40 year period. For row2 in EBR-2 the He/DPA increases from 0.29 to 0.31 over a 40 year period.

The data of Garner and Gelles [33] (reproduced in Figure S3 and Table 1) showed that the swelling rate increases with dose. There are two possible reasons for this non-linearity: (i) there are transmutation reactions creating He that increases in concentration with increasing dose and thus contributes to an increased cavity stabilityand swelling rate; (ii) the microstructure is evolving and the optimum condition for swelling is not achieved until the cavity sink strength is comparable with the dislocation sink strength. In EBR-2 the rate of He production is approximately 0.3 appm He per DPA and the primary DPA rate is approximately $10 \times 10^{-7}$ DPA·s$^{-1}$, corresponding with an FMD rate of either 5 or $10 \times 10^{-8}$ DPA·s$^{-1}$, depending on which model is used for cascade efficiency [57,63]. For the spectrum in question used here (row 2 midplane) there are 4.1 DPA per $10^{22}$ n·cm$^{-2}$ (E > 0.1 MeV). At a neutron fluence of about $20 \times 10^{22}$ n·cm$^{-2}$ (E > 0.1 MeV), corresponding with about 80 DPA, there will only be about 24 appm He generated. This He generation is included in the rate-theory modelling because the He affects the stability of the cavities, especially at higher temperatures (>500 °C). At these low levels of He production, and given the high damage rates in EBR-2, the cavities can be considered voids according to the definition of Bhattacharya and Zinkle [1].

### 5.2. Rate Theory Model

Cavity or void swelling is caused by the formation and growth of voids (empty space) within the microstructure, the excess atoms effectively adding to the total volume of the

material. Whereas it is common for some researchers to refer to voids only [2], a distinction is made here between voids, which are volumes devoid of any atoms within a material, and gas bubbles that are similar to voids but contain gas atoms that help stabilise the void against collapse to a vacancy loop. At a given temperature an equilibrium bubble is one where the driving force for bubble shrinkage by vacancy emission (reduces the surface energy) is balanced by the energy needed to create the vacancy (vacancy formation energy). When He is present, vacancy emission has to overcome the work done against the internal pressure, which is proportional to the ratio of He atoms to vacancies in the cavity (He/V). During irradiation there may be an excess flux of vacancies to bubbles or voids because of the elastic attraction of interstitial point defects with dislocation sinks and there is then a radiation-induced, non-equilibrium, condition that induces more swelling than would be expected from the presence of gaseous atoms alone. One can calculate the swelling using rate theory by considering point defect fluxes (self-interstitials, vacancies, and gas atoms) to the cavities and thus obtain a value for growth (or shrinkage) of the cavities depending on the temperature and neutron flux.

The goal of a swelling model is to calculate the net gain of vacancies in cavities based on the net (bias-driven) vacancy flux and the gain/loss that is a function of the internal pressure (He content). The radiation-induced interstitial and vacancy fluxes to various sinks in the microstructure are determined according to Equations (1)–(5) [67].

$$\phi - \sum_s k_v^2 \, D_v C_v - a C_i C_v = 0 \tag{1}$$

$$\phi - \sum_s k_i^2 \, D_i C_i - a C_i C_v = 0 \tag{2}$$

$$\sum_s k_{i,v}^2 \, D_{i,v} C_{i,v} = \phi \, F(\eta) \tag{3}$$

$$F(\eta) = \frac{2}{\eta} \left\{ (1+\eta)^{0.5} - 1 \right\} \tag{4}$$

$$\eta = \frac{4 \, a \, \phi}{D_v \, D_i \, \sum_s k_v^2 \sum_s k_i^2} \tag{5}$$

where ($\phi$) is the freely-migrating point defect generation rate in atom fraction per second, ($\sum_s k_{i,v}^2$) is the sink strength summed over all sinks (s) and has units of m$^{-2}$, ($D_{i,v}$) is the diffusion coefficient and has units of m$^2 \cdot$s$^{-1}$, and ($C_{i,v}$) is the atom fractional point defect concentration in the diffusive medium. The recombination rate parameter ($a = \frac{r}{a_0^2} \cdot D_i$) where $a_0$ is the lattice parameter, which for a face-centred-cubic (FCC) lattice with a lattice parameter = 0.36 nm, and r is a geometric factor that corresponds with the number of interstitial sites around a vacancy where spontaneous recombination occurs [5]. The recombination rate parameter for stainless steels varies between about $10^{20} \cdot D_i$ [67] and $4 \times 10^{21} \cdot D_i$ [5]. The point defects are designated (i) for interstitial atoms, and (v) for vacancies. The rate of flow of point defects to a void (containing no He) is determined by the concentration difference from the medium mid-way between sinks ($C_\alpha$) and the concentration at the sink ($C_\alpha^{th}$), which is small compared to $C_\alpha$. As $C_\alpha$ consists of thermal- and irradiation-induced vacancies, the vacancy flux term is actually ($C_\alpha + C_\alpha^{th}$) $-$ ($C_\alpha^{th}$). Assuming the thermal equilibrium concentration in the matrix is the same as the concentration at the void surface, i.e., the vacancy formation energy is the same at all sources of vacancies, the thermal vacancy term can often be ignored. By convention the sign of the interstitial flux is positive, and the vacancy flux is negative. For voids and pressurised cavities, excess vacancies may be emitted or absorbed irrespective of irradiation because the thermal vacancy concentration in the matrix may not be the same as the thermal vacancy concentration at the cavity surface.

The net flux of radiation-induced vacancies migrating to cavities is dependent on there being a biased flow of interstitials to sinks other than cavities such as dislocations and grain boundaries. Following the methodology outlined by Wiedersich [71], the approach

adopted here is to consider the flow of point defects to a given sink that is governed by the probability that the sink accepts one type of defect over another. The mechanism of cavity swelling requires that there is a net (biased) flux of interstitial point defects to dislocations. If one approximates cavities and surfaces as neutral sinks with respect to any elastic interaction with self-interstitials and vacancies, the flow of irradiation-induced point defects can be calculated using simple balance equations.

Consider the simple case in the sink-dominated regime (negligible mutual recombination) where the microstructure contains cavities and dislocations only. The balance equations can be formulated as shown in Equations (6) and (7). Representing the flux of point defects (atom fraction per second) to various sinks (s) by $J_s$, where s is c for cavities and d for dislocations, the fluxes to sinks resulting in swelling are given by:

$$J_d = \left[ \frac{(1+b) \cdot \rho_d}{(1+b) \cdot \rho_d + \rho_c} - \frac{\rho_d}{\rho_d + \rho_c} \right] \cdot \Phi \qquad (6)$$

$$J_c = \left[ \frac{\rho_c}{(1+b) \cdot \rho_d + \rho_c} - \frac{\rho_c}{\rho_d + \rho_c} \right] \cdot \Phi \qquad (7)$$

The bias, *b*, represents the difference between the elastic interaction of dislocations with interstitial compared with vacancy point defects. The bias parameter (b) is the probability that interstitial point defects rather than vacancy point defects will be absorbed at a dislocation because of the elastic interaction difference [67]. The terms $\rho_d$ and $\rho_c$ represent the unbiased sink strengths (no elastic interaction) for dislocations and cavities [67], and $\Phi$ is the freely migrating point defect production rate, dpa·s$^{-1}$. The unbiased dislocation sink strength is given by the dislocation density, $\rho_d$, in units of m$^{-2}$, and the unbiased cavity sink strength is denoted $\rho_c$ and has a value of $4\pi r_c N$, also in units of m$^{-2}$, where $r_c$ is the cavity radius in metres (m) and N is the number density in m$^{-3}$ [67].

Based on reaction-rate kinetics [5], the sink strength for dislocations is denoted,

$$k_{i,v}^2 = \frac{1}{L_{i,v}^2} = z_{i,v} \rho_d \qquad (8)$$

where *L* is the average distance that a given point defect travels before being captured by a dislocation, and $\rho_d$ is the dislocation density.

The interstitial bias for dislocations is given by [67],

$$b = \frac{Z_i - Z_v}{Z_i} \qquad (9)$$

By solving diffusion equations with cylindrical geometry Heald and Speight [67] showed that the rate constant for point defect interactions with dislocations is

$$z_{i,v} = \frac{2\pi}{\ln \frac{2R}{l_{i,v}}} \qquad (10)$$

where *R* is the mean distance between dislocations and $l_{i,v}$ is the radius of an effective trapping cylinder around the dislocation core. The value of $l_{i,v}$ is determined from the elastic strain field interactions between the dislocation and respective point defects. It is different from $L_{i,v}$, which is the average distance a point defect travels before being captured at a given sink.

Using the same generic parameters applied to 316 SS by Heald and Speight [67] the bias parameter (b) for different dislocation densities as a function of temperature vary between 0.329 at 650 °C and 0.364 at 400 °C (see Figure S2 in supplementary materials).

An illustration of the dependence of radiation-induced swelling rate on dislocation and cavity sink strengths for voids using Equations (6) and (7) is shown in Figure 9 assuming that one is in a sink-dominated regime and mutual recombination is negligible. Note that, in this example, the nominal cascade efficiency is 1% for a primary displaced

atom production rate of $10^{-7}$ DPA·s$^{-1}$. The plot shows that, as the cavity density (sink strength) evolves, the swelling rate passes through a peak value when the net partitioning of point defects to the different sink types is optimal.

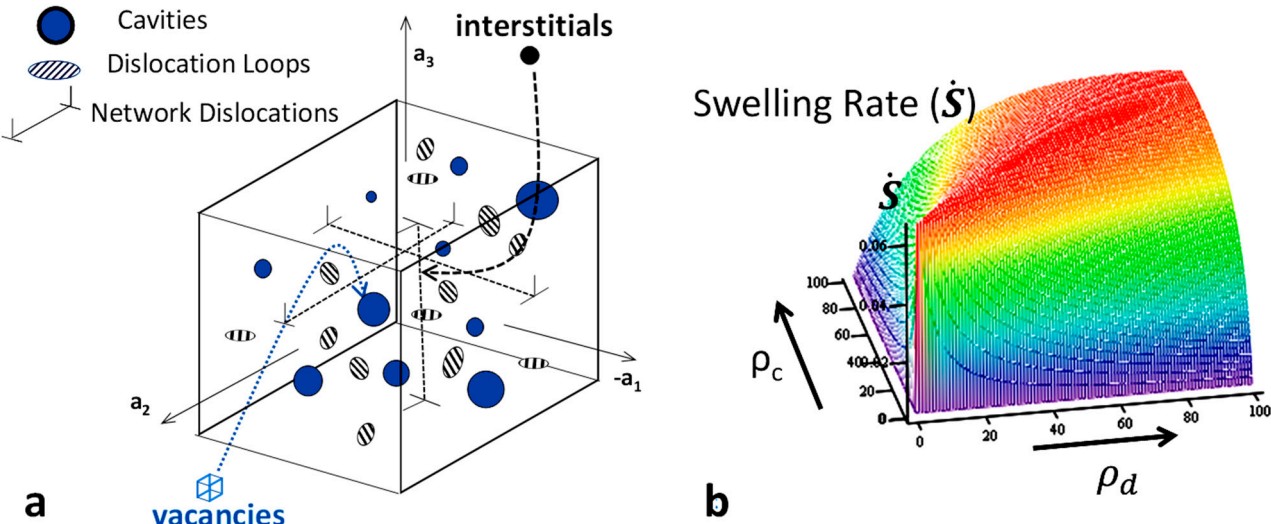

**Figure 9.** (**a**) Schematic of microstructure and (**b**) rate theory output as a colour contour plot, showing irradiation swelling rate ($\dot{S}$) as a function of the relative dislocation ($\rho_d$) and cavity/void ($\rho_c$) sink densities/strengths in units of m$^{-2}$. The swelling rate is small when $\rho_d$ and $\rho_c$ are very different. The volume swelling rate (% per unit DPA) is a maximum (red shading) when $\sqrt{1+b} \cdot \rho_d = \rho_c$, where $b$ is governed by the size-effect interstitial bias for dislocations. For this calculation the cascade damage efficiency is a nominal 1% and the dislocation bias for interstitials ($b$) = 0.3. In this case the maximum swelling is observed when $\rho_c = 1.14 \cdot \rho_d$.

As one can see, for a given dislocation density, as the cavities nucleate and grow, the cavity sink strength increases and the swelling rate will increase accordingly. As the cavity sink strength increases, the partitioning of vacancies to cavities and interstitials to dislocations will reach a maximum value. This occurs when the ratio of the unbiased vacancy and interstitial sink strengths is equal to $\sqrt{1+b}$, where $b$ is the bias differential for interstitial point defect diffusion to dislocations due to an elastic interaction, i.e., when the unbiased cavity sink strength is $\sqrt{1+b}$ times the unbiased dislocation sink strength. Logically, swelling cannot increase indefinitely. Whether by increasing the number density of cavities or by an increase in their size, an evolving cavity structure will compete for, and absorb, more interstitials as the cavity sink strength increases. Recombination, other than the mutual recombination from chance encounters between interstitials and vacancy point defects, effectively increases when interstitials and vacancies annihilate at the same sink. This type of recombination is largest when the sink strengths differ from the optimum partitioning condition, hence the lower rate of swelling. The interstitial bias is a factor that effectively increases the sink strength of a given sink for a particular point defect; in this case the bias of dislocations for attracting interstitial point defects [67].

Inclusion of grain boundary sinks is achieved using the sink strength formulae developed by Heald and Harbottle [72]. In their work the grain boundary sink strength is shown to be a function of the average sink strength in the grain interior. Effectively, for a point defect in the vicinity of a grain boundary the distance travelled before encountering the boundary, as opposed to any other sink, is equal to the average value of L within the grain interior [73]. When coupled with the grain boundary density the expression for grain boundary sink strength for cases where the grain diameters are large compared with the average diffusion length in the grain interior is,

$$\left(k_{i,v}^{gb}\right)^2 = \frac{6 \cdot \sum_s k_{i,v}}{d} \tag{11}$$

where $d$ is the mean grain diameter.

Factoring in the role of grain boundaries as sinks the total sink strength is used to determine the steady state point defect concentrations ($C_{i,v}$) according to equation (3) and that portion of the point defect fluxes to cavity sinks then becomes,

$$J_c = k_i^2 D_i C_i - k_v^2 D_v C_v \tag{12}$$

where, for cavities of radius $r_c$ and number density N, $k_i^2 = k_v^2 = 4 \cdot \pi \cdot r_c \cdot N$ [67], or

$$J_c = \left[ \begin{array}{c} \dfrac{4 \cdot \pi \cdot N \cdot r_c \cdot F(\eta)}{(1+b) \cdot \rho_d + 4 \cdot \pi \cdot N \cdot r_c + \frac{6 \cdot \sqrt{(1+b) \cdot \rho_d + 4 \cdot \pi \cdot N \cdot r_c}}{d}} \\ - \dfrac{4 \cdot \pi \cdot N \cdot r_c \cdot F(\eta)}{\rho_d + 4 \cdot \pi \cdot N \cdot r_c + \frac{6 \cdot \sqrt{\rho_d + 4 \cdot \pi \cdot N \cdot r_c}}{d}} \end{array} \right] \cdot \Phi \tag{13}$$

As the microstructure evolves the number density and/or size of cavities increases monotonically with increasing dose, especially when He is also being generated. For a material containing a high dislocation density (from cold-working or dislocation loop formation), as the cavity number densities and sizes increase the swelling rate will also increase up to a point where the cavity sink strength is comparable with the dislocation sink strength [26,27]. At this point the swelling rate from the irradiation-induced vacancy and interstitial point defects will have reached its maximum value, i.e., for the case of voids with no He atoms in the cavities. If He is being continuously generated, the He itself has an effect on the propensity for vacancies to annihilate at cavities; the He, in effect, increases the capacity of the cavities to absorb more vacancies. Any rate theory model must therefore factor in the effect of the arrival of He at an evolving sink structure.

The net vacancy flux to/from He-containing cavities (radius $r_c$) is a combination of an irradiation term ($J_c$), plus the flux resulting from thermal vacancies in the continuum, minus an emission term [40,67],

$$\text{Net vacancy flux to cavity} = -\left(J_c - k_v^2 D_v \left(C_v^{th} - C_v^{emit}\right)\right) \tag{14}$$

where the concentrations are given in fractional units (point defects per atom),

$$C_v^{th} = exp\left(-\frac{E_f}{k \cdot T}\right) \tag{15}$$

$$C_v^{emit} = exp\left(\frac{-E_f - P_c \cdot \Omega + \frac{2 \cdot \gamma \cdot \Omega}{r_c}}{k \cdot T}\right) \tag{16}$$

$J_c$ is the net radiation-induced flux of vacancy point defects to the cavity. It is negative with respect to vacancies so $-J_c$ represents the vacancy flux expressed as a positive number. $C_v^{th}$ is the vacancy thermal equilibrium concentration and $C_v^{emit}$ is the term describing the emission of vacancies from a pressurised cavity based on the balance of surface energy, pressure and the vacancy formation energy, $E_f^v$. The thermal equilibrium interstitial concentration and the corresponding thermal emission for interstitials are ignored because they are so small relative to the vacancy values. $P_c$ is the internal pressure due to He, $\Omega$ is the atomic volume, $r_c$ is the cavity radius and $D_v$ is the vacancy diffusion coefficient. The volume swelling $\left(\frac{\Delta V}{V}\right)$ is simply given by the net fractional vacancy accumulation in cavities. For most cases during irradiation the thermal vacancy concentration can be ignored because it is many orders of magnitude smaller than the radiation-induced steady-state concentration. The thermal term is included in this case to show that when the material is no longer being irradiated, the cavity can either shrink or grow depending on whether the term $C_v^{emit}$ is greater than or less than the thermal equilibrium vacancy concentration, i.e., whether $P_c$ is greater or less than $\frac{2 \cdot \gamma}{r_c}$. When $P_c = \frac{2 \cdot \gamma}{r_c}$ the net emission of vacancies from the cavity due to statistical fluctuations dictated by the formation energy is balanced

by the thermal equilibrium vacancy concentration in the matrix that is maintained by all other sources, for example dislocations and grains boundaries. Under these conditions the cavity is termed an equilibrium bubble. In essence, at equilibrium the emission term is balanced by an equal and opposite absorption term. It should be noted here that there was a typographical error in the equation that is equivalent to Equation (14) and reported previously [51]. Equation (5) in [51] has the incorrect sign for the vacancy emission term—it should have been negative rather than positive. The error was typographical only and did not affect the calculations in that publication.

Modelling the evolution of the cavity structure is complicated and subject to large uncertainties in the values of the parameters. The sink evolution can be assessed from experimental data and one can thus project how the swelling rate will change as the cavity structure evolves. Swelling is largely controlled by the migration of freely migrating point defects. At temperatures where vacancy point defects are mobile one can calculate the swelling rate using rate theory for a given microstructure, temperature, and production rate of FMDs. The FMDs are those point defects that survive spontaneous recombination in collision cascades [57–66]. The FMDs then migrate by solid state diffusion to various sinks. If both types of point defects (interstitial and vacancy) migrate to the same type of sink, then there is no net change in swelling. Swelling can only occur if there is a net flux of interstitials to biased sinks such as dislocations and a net flux of vacancies to neutral sinks such as cavities. The effect of He is important because He stabilises cavities, thus enhancing swelling. Nucleation aside, as we shall see in Section 6, the stabilisation of the cavities is most important at high temperatures. One can assume that, after a dose of about 10 DPA, the dislocation structure is more-or-less constant as evidenced by the yield strength [2], see Figure 5. The dislocation structure is a combination of the original network dislocations introduced by fabrication and dislocation loops caused by interstitial point defect clustering. The network dislocations can recover, and new dislocation loops can be created by point defect clustering and growth depending on the temperature and damage rate. Empirically, the dislocation structure evolves to a more-or-less constant state even though the cavities may continue evolving. The swelling can be calculated assuming a constant dislocation structure and a cavity structure that is evolving from the absorption of vacancies and also He atoms. The cavity evolution rate must eventually drop to a minimum value as the cavity sink strength (based on size and number density) reaches a sufficiently large value that the cavities are dominant sinks for both interstitials and vacancies.

The swelling rate can be determined using Equations (8)–(16). The relevant microstructural and parameter inputs are listed in Table 3 [9,15,32,39,74,75]. Note that the He diffusion coefficient of 0.14 eV is that of the interstitial He atom [74] rather than the He-V complex, the latter having a much higher migration energy of 0.8 eV [76]. Note also that many authors use a recombination number of r = 48 as recommended by Olander, see page 208 in [5]. However, Olander also suggests that a much higher number (r = 500) may be warranted. Calculations have been performed using both parameters. The swelling is determined iteratively, by taking account the volume of the cavity structure and the arrival rate of point defects (including He atoms) at each stage of the swelling computation as shown by the algorithm in Figure 10. The pressure at each time step is calculated using the Carnahan-Starling equation of state assuming that the [He] is uniformly dispersed in all cavities [77]. The time-steps are broken down into 100 equal parts each corresponding with a duration of 0.1 years of operation. The computation therefore occurs over a period of 10 years corresponding to a final primary damage after 10 years of 130 DPA for the peak flux in row 8 (D5) and 335 DPA for row 2 (midplane). The grain size is nominal in each case (10 μm diameter). Note that in these calculations the role of boron in producing He has not been included. Firstly, because the amount of B is unknown and secondly, because B is likely only present at low concentrations [60], and the amount of He produced is negligible compared to that produced by Fe, Cr and Ni, especially in a fast reactor.

**Table 3.** Parameters used in calculating volume swelling in stainless steel. Note the pre-exponential factor for interstitial He is assumed the same as for a self-interstitial atom.

| Parameter | Value | Reference |
|---|---|---|
| Vacancy migration energy | 1.4 eV | [32] |
| Vacancy formation energy | 1.6 eV | [32] |
| Interstitial migration energy | 0.15 eV | [32] |
| Interstitial formation energy | 3 eV | [32] |
| Pre-exponential factor, $D_0^i$ | $12 \times 10^{-6}$ | [32] |
| Pre-exponential factor, $D_0^v$ | $6 \times 10^{-6}$ | [32] |
| Recombination factor | 50 or 500 | [15] |
| He Interstitial Migration Energy | 0.14 eV | [74] |
| He Interstitial Pre-exponential factor, $D_0^{He}$ | $12 \times 10^{-6}$ | [32] |
| Lattice Parameter, $a_0$ | 0.363 nm | [39] |
| Surface Energy, S | 1 or 2 J·m$^{-2}$ | [9] or [75] |

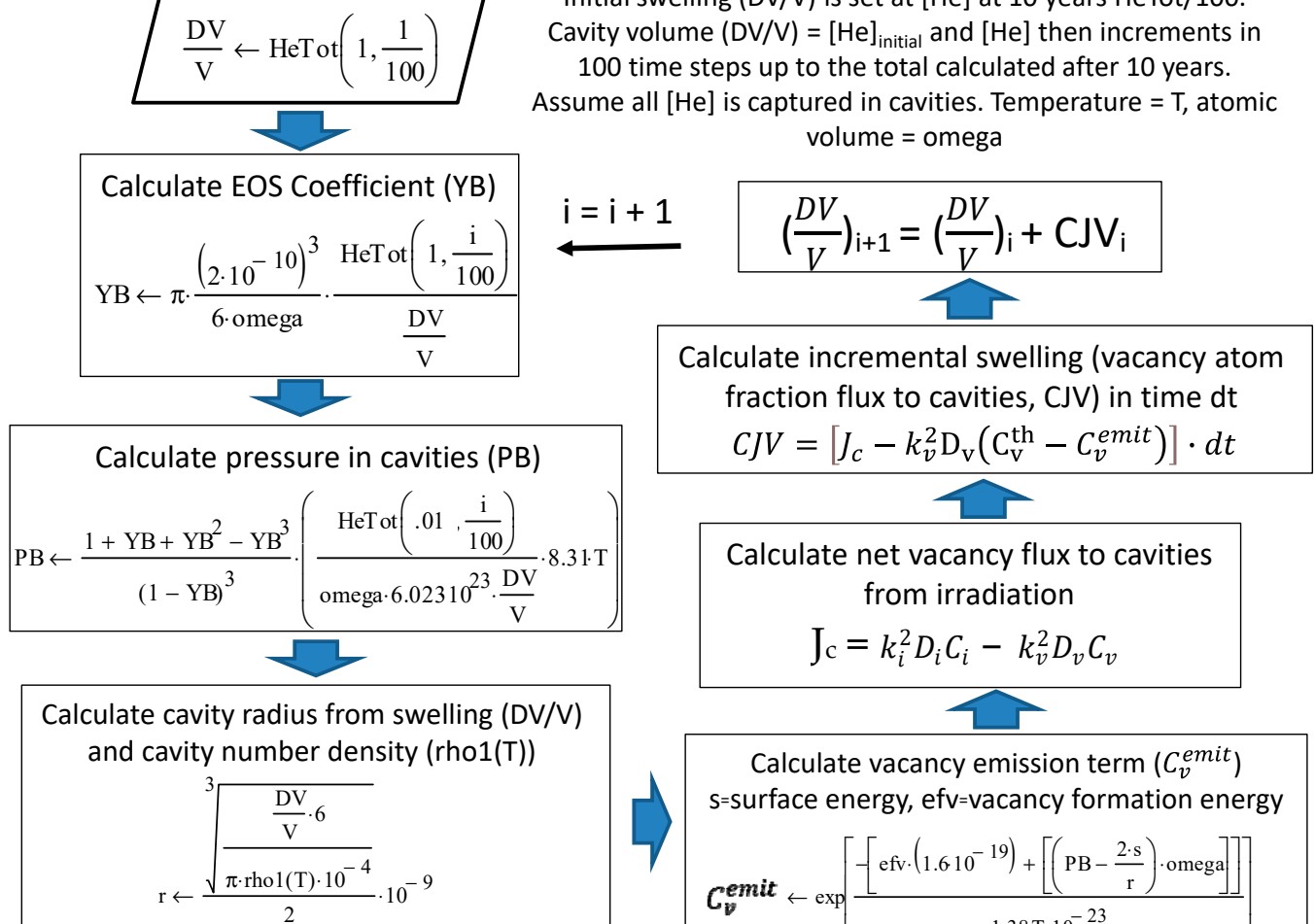

**Figure 10.** Algorithm for computing irradiation-induced swelling (row 8 case). The initial value for the seed cavity volume is the value of accumulated [He] in the first time-step (0.1 years) assuming all the He is captured by vacancies.

## 6. Results

### 6.1. Effect of Helium on Swelling Rate

The effect of increasing the ratio of He atoms to vacancies within the cavities as a function of temperature at high doses can be demonstrated using a nominal set of microstructure parameters, i.e., fixed grain diameter of 10,000 nm, dislocation density according to that shown in Figure 7, and a fixed cavity sink strength (cavity number density of $10^{22}$ m$^{-3}$, cavity diameters of 5 nm). The effect of varying the He/V ratio is illustrated in Figure 11 for two different recombination factors (r = 50 and r = 500). The FMD rates are nominally $10^{-8}$ s$^{-1}$ and $10^{-7}$ s$^{-1}$. One can see that, whereas low temperature swelling behaviour is relatively insensitive to the He/V ratio, at high temperatures the effect of having a high He atom fraction in the cavities is to greatly enhance the swelling rate. The low value of the [He]/vacancy ratio corresponding with Figure 11a is the steady-state value at high doses obtained from the model (see Section 7). At such a low value the cavity is effectively a void [1]. The plots show that, as one might expect, increasing the magnitude of the recombination parameter, r, from 50 to 500 shifts the transition between the recombination-dominated and sink-dominated regimes to higher temperatures. More importantly, Figure 11 shows that He is important in enhancing swelling at high temperatures.

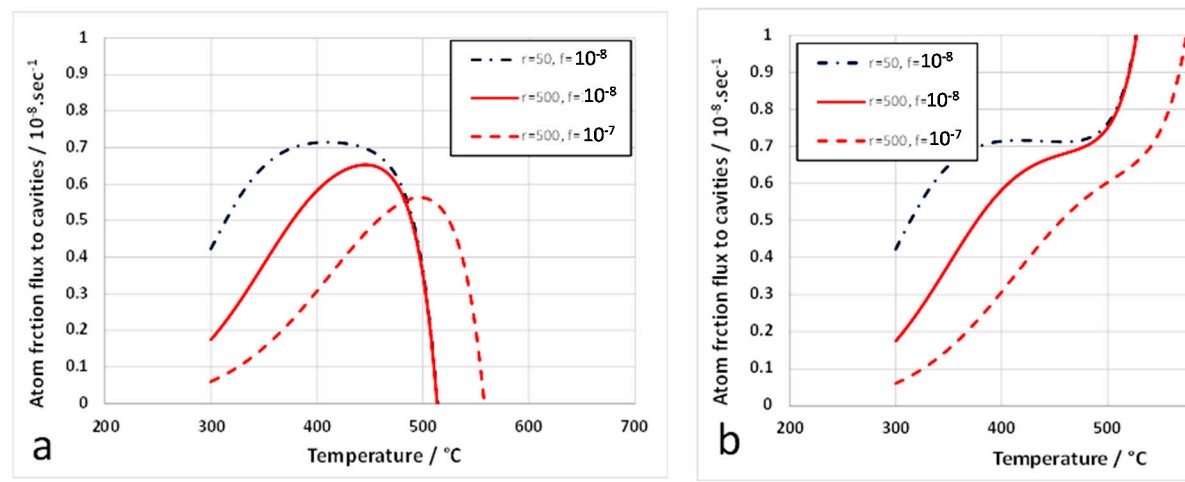

**Figure 11.** Atom fraction vacancy fluxes to cavities after 10 DPA (saturated dislocation densities) for a cavity number density of $10^{22}$ m$^{-3}$, cavity diameters of 5 nm and two different He/V concentrations in the cavities: (**a**) He/V = 0.0015; (**b**) He/V = 1. The FMD atom fraction production rates are f = $10^{-8}$ s$^{-1}$ and f = $10^{-7}$ s$^{-1}$ for recombination factors of r = 50 and r = 500 (see text). The cavity surface energy = 2 J·m$^{-2}$.

### 6.2. Effect of FMD Production Rate on Swelling Rate

The ability of the model to capture the effect of varying the damage rate is illustrated in Figure 12. Data for swelling of 316 SS in EBR-2 at 400 °C [78] indicates that increasing the damage rate increases the incubation dose for the onset of high swelling rates. Applying the model algorithm as outlined in Figure 10, for high and low FMD rates, there is a shift in the onset of high swelling rates to higher doses when the swelling rate is high. The data and model predictions are compared in Figure 12a,b, respectively. The high DPA rate data in Figure 12b use the calculated He and DPA rate corresponding to row 2 in EBR-2. The low DPA rate is simply an order of magnitude smaller. There are two cases: one where both the DPA and [He] production rate are an order of magnitude smaller, and one where only the DPA rate is reduced relative to row 2 conditions in EBR-2. The effect of decreasing the DPA rate on the shift in the onset is most important. The shift is relatively insensitive to a comparable change in the [He] production rate at this temperature.

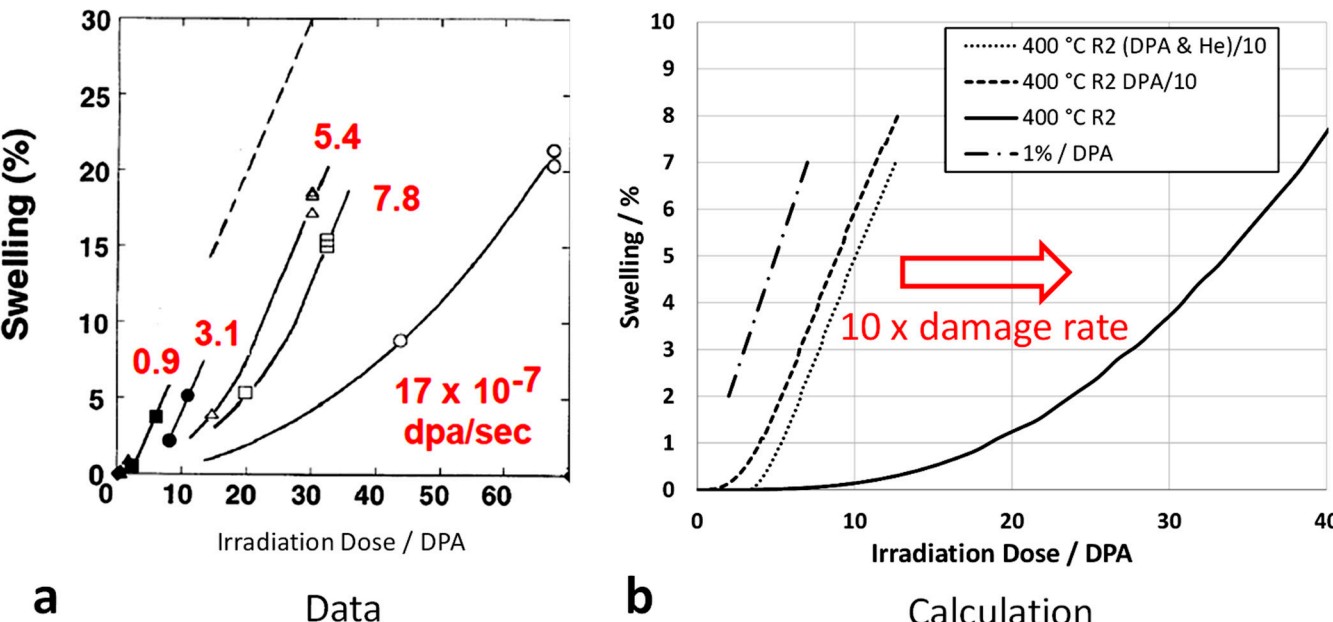

**Figure 12.** (**a**) Swelling as a function of primary DPA for Fe-15Cr-16Ni irradiated in EBR-2 at 400 C; (**b**) Predicted swelling for 316 SS irradiated in EBR-2 at 400 C at two different DPA rates. The solid curve corresponds with then conditions established for 316 SS irradiated in row 2 at 400 °C. The dashed curves show how the onset of high swelling rates occurs at a smaller incubation dose when the damage rate is reduced by a factor of 10, closer to power reactor conditions.

### 6.3. Effect of Temperature on Swelling

The intent of the current modelling effort is to match the swelling data given in Table 1 (Figure S3) using microstructure, He production and atomic displacement rates that, as closely as possible, apply to the data. Whereas the damage rate and temperatures can be reproduced adequately, the microstructure is unknown. Some assumptions can be made regarding grain size based on what is common for engineering alloys. A value of 10,000 nm has been chosen, which is at the low end of expected for most engineering alloys. Irrespective, the model is not sensitive to grain size until it is comparable to the mean diffusion length, which in this case is <100 nm at all temperatures. The justifications for the choice of cavity number densities and dislocation densities have been described in Section 4.

The model algorithm shown in Figure 10 has been applied using the microstructure inputs as described in Section 3, the equations of Section 5, and the parameters given in Table 3. The swelling has been calculated for each temperature using: (i) the dislocation densities shown in Figure 7; (ii) the two cavity number densities shown in Figure 8; (iii) recombination parameters of r = 50 and r = 500; and (iv) surface energies of 1 and 2 J·m$^{-2}$. The calculation shows reasonable agreement at high temperatures but poor agreement at lower temperatures with respect to the incubation dose for the onset of high swelling rates. This discrepancy can be corrected to some extent by applying a dose shift to the low temperature data below 500 °C of 0.5 DPA per degree for each degree below 500 °C. With the inclusion of a dose incubation shift for temperatures < 500 °C the best fit to the data is obtained using a recombination factor = 500 and a surface energy of 1 J·m$^{-1}$, Figure 13.

Swelling is insensitive to the surface energy at low temperatures but the surface energy has a small effect on swelling at high temperatures (650 °C). The surface energy is only weakly dependent on temperature, varying by 0.25 J·m$^{-2}$ between 400 °C and 650 °C [12], and is often assumed to be constant for the purposes of modelling in many publications.

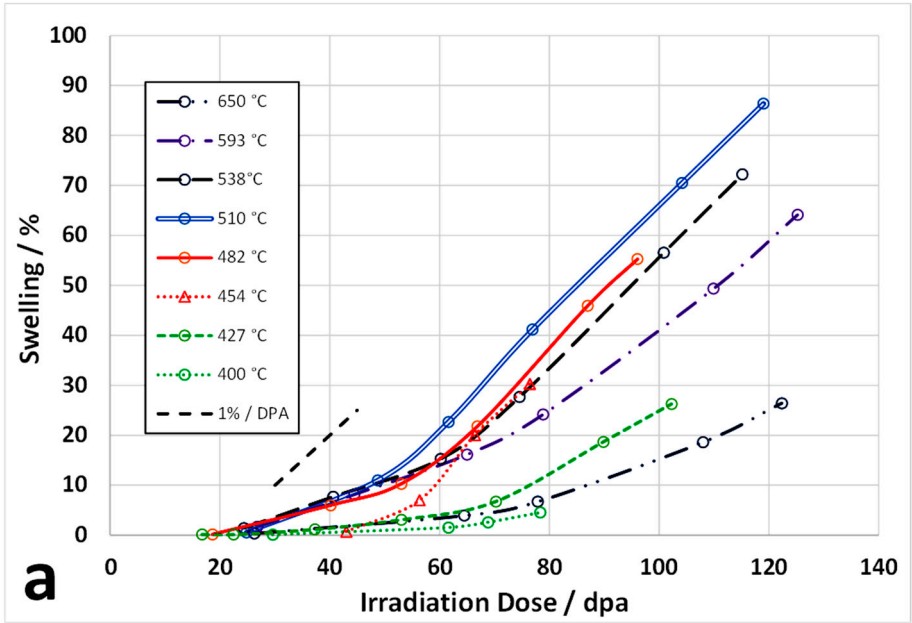

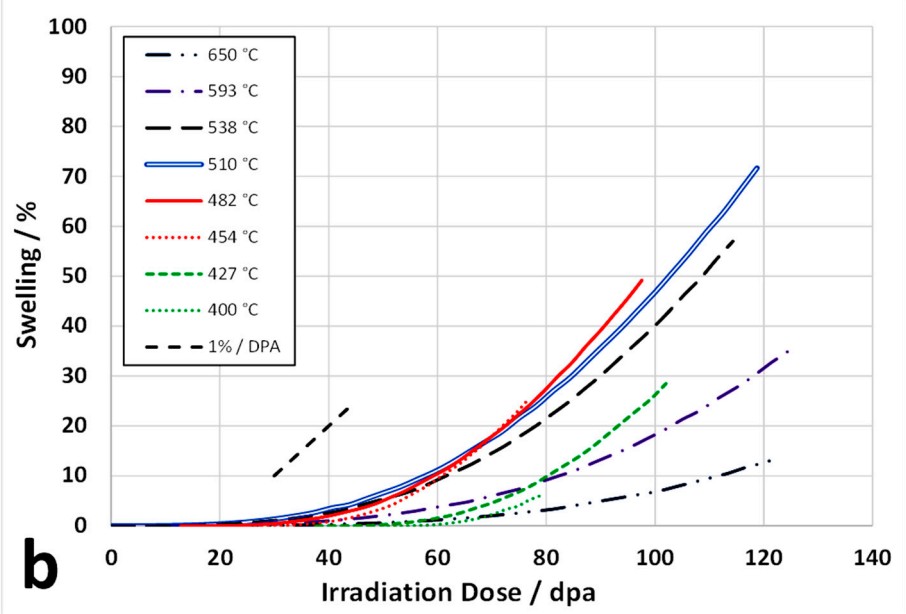

**Figure 13.** (**a**) measured swelling as a function of dose at various temperatures (see Table 1); (**b**) predicted swelling using rate-theory model with the dislocation density as shown in Figure 7, a recombination parameter (r) = 500, a surface energy of 1 J·m$^{-2}$ and an incubation dose shift of 0.5 DPA per degree below 500 °C.

The justification for applying an empirical dose shift as described is based on evidence that the incubation dose for the onset of high swelling rates is not only determined by the damage rate (as shown in Figure 12) but also by the dislocation density [2,68]. There is no provision in the model to include an incubation term based on nucleation of the cavities; the assumption is that they form nuclei at a low dose (<3 DPA) and the cavity number density remains constant thereafter. Empirically one must apply a temperature dependent shift to better match the swelling measurements that may be related to the dislocation density, which is higher at lower temperatures (see Figures 5–7).

Another way to compare measured and predicted swelling is to calculate the swelling rates at the maximum doses of the measured swelling data shown in Figure 13a. The corresponding measured and calculated swelling rates are shown in Figure 14. A comparison

of the calculated swelling using recombination factors of r = 50 and r = 500 for an FMD production calculated using the Kwon and Motta [57] formula and the formula derived from the data of Okamoto et al. [63,65], for the case where the surface energy = 2 J·m$^{-2}$, is given in supplementary materials, Figure S4.

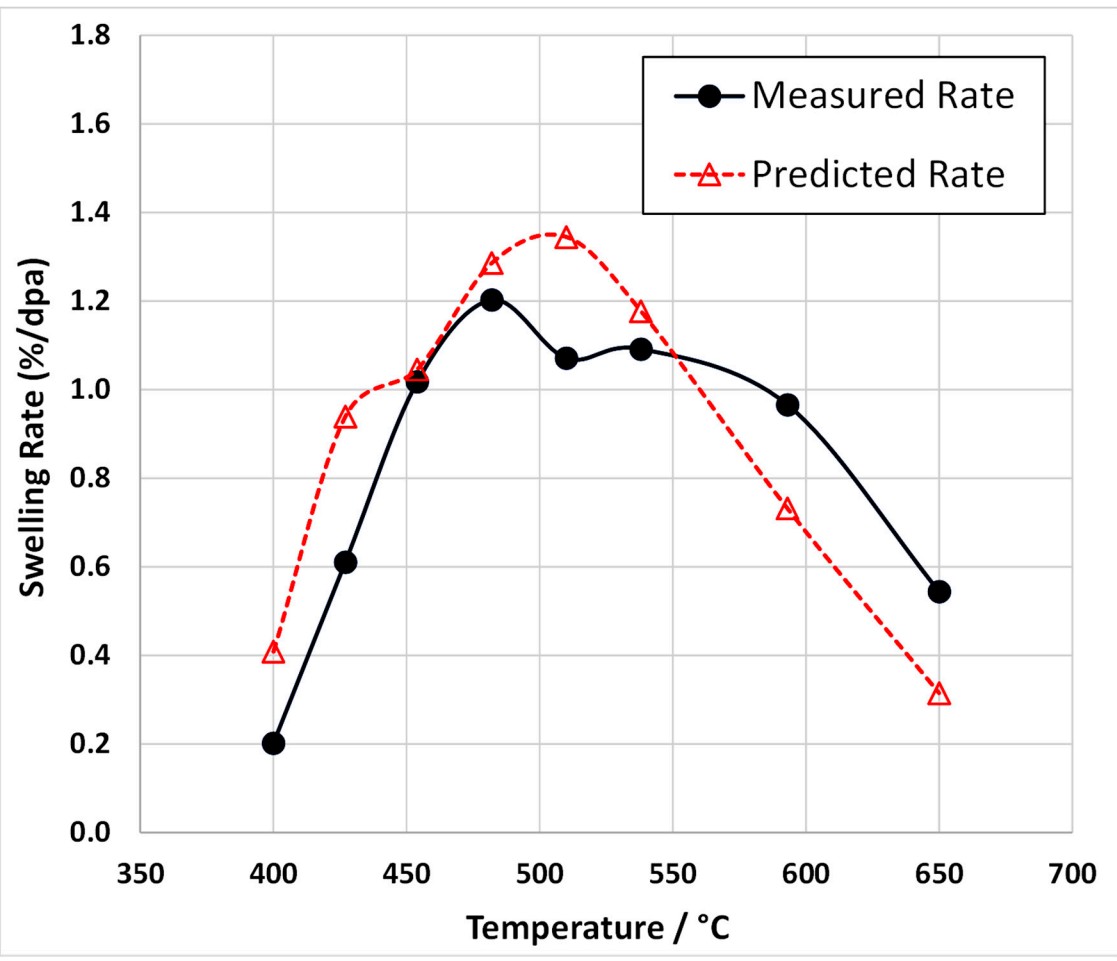

**Figure 14.** Measured and predicted volume swelling rates at the maximum doses shown in Figure 13 for cold-worked 316 stainless steel irradiated in EBR-2, row 2 at various temperatures and doses.

Whereas the low temperature data can be improved by applying a dose shift, the model at high temperatures is improved and brought closer to the measured data by using a surface energy of 1 J·m$^{-2}$ as opposed to 2 J·m$^{-2}$, Figure S5 in supplementary materials. The plots in Figure S5 also show that choosing the lower of the two cavity density curves, shown in Figure 8b, and the Okamoto et al. fit for FMDs, shown in Figure 3, give the best fit.

## 7. Discussion

The radiation response (swelling) of any material is ultimately a function of the microstructure that dictates the relative sink strengths for vacancy and interstitial point defects for different microstructural features, be they cavities (voids or gas bubbles), network dislocations, dislocation loops, precipitates or grain boundaries. One cannot simply attribute the propensity for void swelling to any single microstructural variable except in a strictly controlled experiment. Variables such as temperature and dose also affect the microstructure and one must take into account the microstructural changes when applying a model to predict the swelling as a function of temperature, dose and dose rate. The measurement of microstructure is itself fraught with difficulties and uncertainties. This paper has been

aimed at exploring the application of a rate theory model to swelling data from samples irradiated in EBR-2 Row 2 by employing best estimates of material parameters while at the same time using best estimates of microstructure inputs to the model. The neutron flux for the data in question varied from 65% of the maximum value [33] and the effect of the variation in radiation damage rates on swelling at different temperatures was not large. In a separate experiment 78] from EBR-2, the effect of a 10-fold change in dose rate was studied for swelling of a ternary austenitic FeCrNi alloy at 400 °C. Using the model described in Section 5 the measured flux dependency could be emulated, as shown in Figure 12. Because damage rate effects and irradiation temperature are intimately related, a fuller validation of the model vis-à-vis damage rate dependence is not possible with the current data set.

The science surrounding the calculation of primary atomic displacements is mature and any refinements on primary damage production cross-sections tend to be small. The SPECTER code [34] has been increasingly used and can be considered an industry standard. With that being said, rate theory models rely, not on the primary displacements, but on those defects that are freely migrating because mass transport (diffusion) of the mobile point defects is a key component. The choice of a value for FMD fraction has been subject to considerable uncertainty and values ranging from 2% to 25% have been reported [14]. New data from Faulkner et al. [58] indicate that the low values of FMD fraction that have been adopted by many researchers may not be justified. In this study the FMD fraction was determined from data by Okamoto et al. [63,65]. For the spectrum in EBR-2 row2 the FMD fraction was determined to be about 10%. This value is consistent with other values derived from molecular dynamics [58]. One can always argue that other (unknown) factors may be responsible for the high apparent swelling rates but, in the absence of any data to the contrary, an FMD fraction between 10% for fast reactors and 15% for power reactors, based on both experimental data [63] and molecular dynamics [58], seems to be appropriate.

One other key component of a void swelling model is the determination of the pressure that stabilises the cavity. Using the Carnahan-Starling equation of state to calculate the cavity pressure [78], the cavity growth rates as a function of cavity diameter can be determined using the algorithm shown in Figure 10. Maintaining a constant cavity number density (determined by nucleation in the early stages of irradiation), the cavity growth rate has been computed considering the effects of varying the He content and the cavity volume on the internal pressure and thus the propensity for emitting (or absorbing) vacancy point defects. Apart from assuming a constant cavity number density with increasing neutron dose in the model, the choice of microstructure variables have been described and justified in Section 3, and the main material parameter variables have been listed in Table 2. One other factor that is subject to some uncertainty is the choice of the recombination constant (r), see Section 5. Many researchers use a value of 48 that is recommended by Olander for FCC materials [63]. However, Olander also recommends a factor of 500. The main effect of a higher recombination number in a model calculation is a reduction in swelling at low temperatures. Using a recombination factor (r) equal to 500 gives a better fit to the measured data (Figure S4).

Even though the best estimates of microstructure and FMD production have been applied, there are still some areas where the model cannot adequately predict the behaviour at all temperatures. The best matching shape for the predicted swelling rate as a function of temperature is obtained by using the assumed dislocation density as a function of temperature shown in Figure 7, coupled with a cavity number density as a function of temperature (corresponding with the lower dashed line) in Figure 8b [1]. From this base, better agreement with the swelling-dose curves can be obtained by assuming that there is an incubation for cavity formation at temperatures up to 500 °C. The best fit to the experimental data can be obtained by applying a dose shift to the calculated values that is 0.5 DPA per °C below 500 °C. The results showing the best agreement between the model and measurements are shown in Figures 13 and 14. Bearing in mind that there was

an empirical dose shift applied to low temperature data there is reasonable agreement between the model and the measured data. Applying the empirical temperature shift is justified by the known effect that a higher dislocation density at low temperatures has on the incubation period for the onset of high swelling rates [2,68]. This delay or incubation dose for the onset of swelling that is observed for high dislocation densities may be because of trapping of He by dislocations [1]. The swelling rate is ultimately dictated by the balance between dislocation and cavity sink strengths. The inability of the model to capture the longer incubation dose at low temperatures reflects an inadequate depiction of the cavity evolution. A future refinement of the model will require accurate observations and measurements on the microstructure as it evolves with increasing neutron dose.

Given that one can model the swelling of irradiated stainless steel, albeit semi-empirically, the amount of predicted swelling shows that the cavities are non-equilibrium and under-pressurised during irradiation in EBR-2. The plot in Figure 15 gives an idea of how the activation energy for emission of a vacancy from a cavity (2 nm in diameter) varies as a function of the ratio of He atoms and vacancies in the cavity (the He/V ratio) at different temperatures. The plot illustrates when a cavity is at thermal equilibrium, i.e., where the pressure curves cross the surface energy dashed line. Values of the pressure curves below the horizontal dashed line mean that the cavity will shrink (under-pressurised) and vice versa for the values above the dashed line. In this example thermal equilibrium is achieved for a 2 nm diameter cavity that has a He/V ratio = 1 at 300 °C. In the absence of any irradiating flux the cavity will shrink for all conditions lying below the horizontal dashed line and grow for the conditions that exist above the horizontal dashed line.

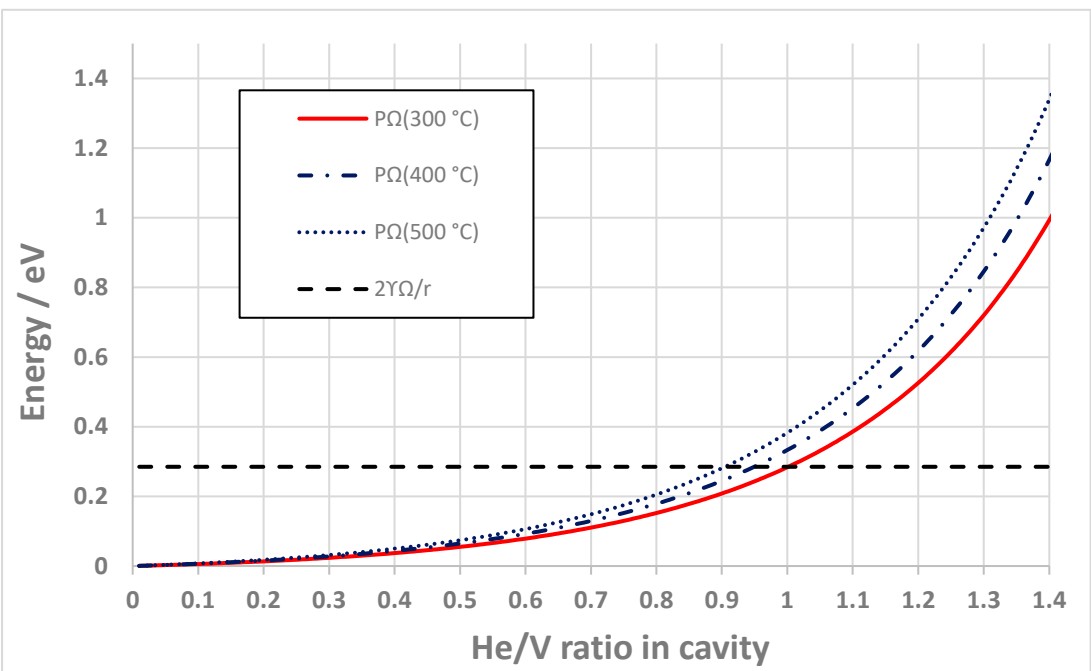

**Figure 15.** Activation energy for vacancy emission from cavities. The plots show the relative stability of He-containing cavities as a function of the ratio of He atoms and vacancies (cavity volume). The work done against pressure due to the emission of one vacancy ($P\Omega$) is compared with the decrease in surface energy ($2\gamma\Omega/r$) where $\Omega$ is the atomic volume, for a cavity radius r = 1 nm and surface energy = 2 J·m$^{-2}$, at various temperatures.

When calculating the swelling evolution as a function of DPA the main parameters that affect the swelling rate, apart from the point defect flux arising from irradiation, are the cavity pressure (dictated by the He/V ratio), the cavity diameter and the combined effect of the pressure and diameter on the activation energy for vacancy emission. These values are plotted in Figures 16 and 17 for the evolving cavity microstructure described by the model. Note that there has been no dose shift applied in these cases.

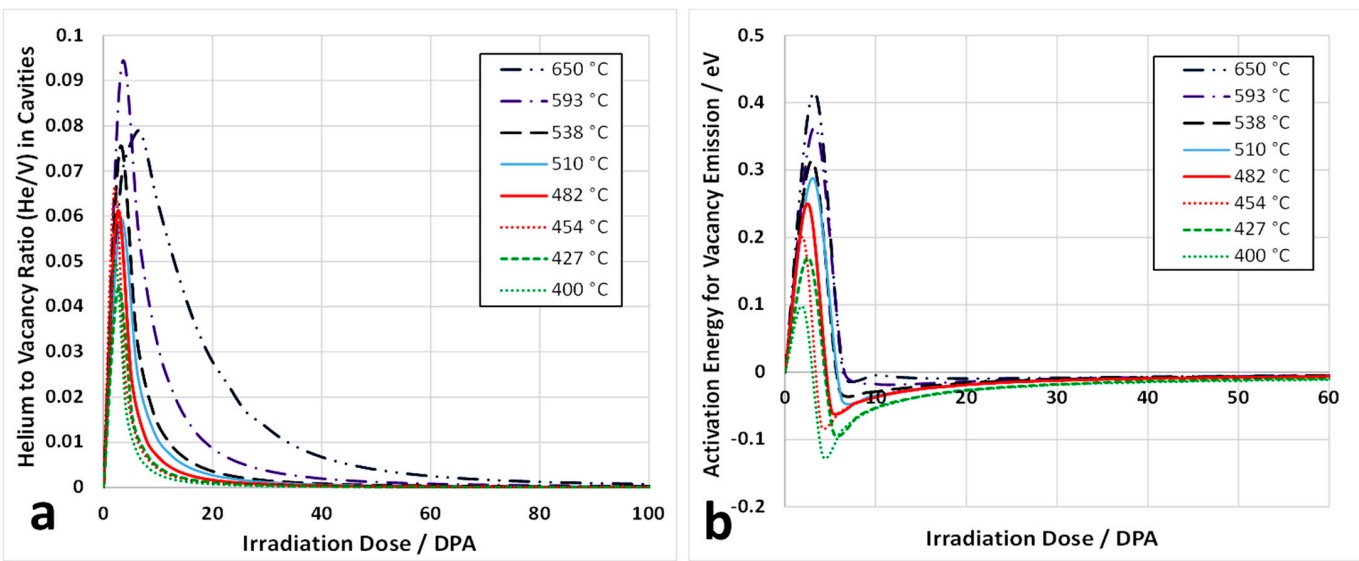

**Figure 16.** Calculated evolution of cavity properties during irradiation in Row 2 of EBR-2: (**a**) He/V ratios as a function of DPA: (**b**) Activation energies (AE) for vacancy emission = $p\Omega - \frac{2\gamma}{r}$ as a function of DPA. Positive values mean vacancy emission will be energetically unfavourable and vice-versa for negative values.

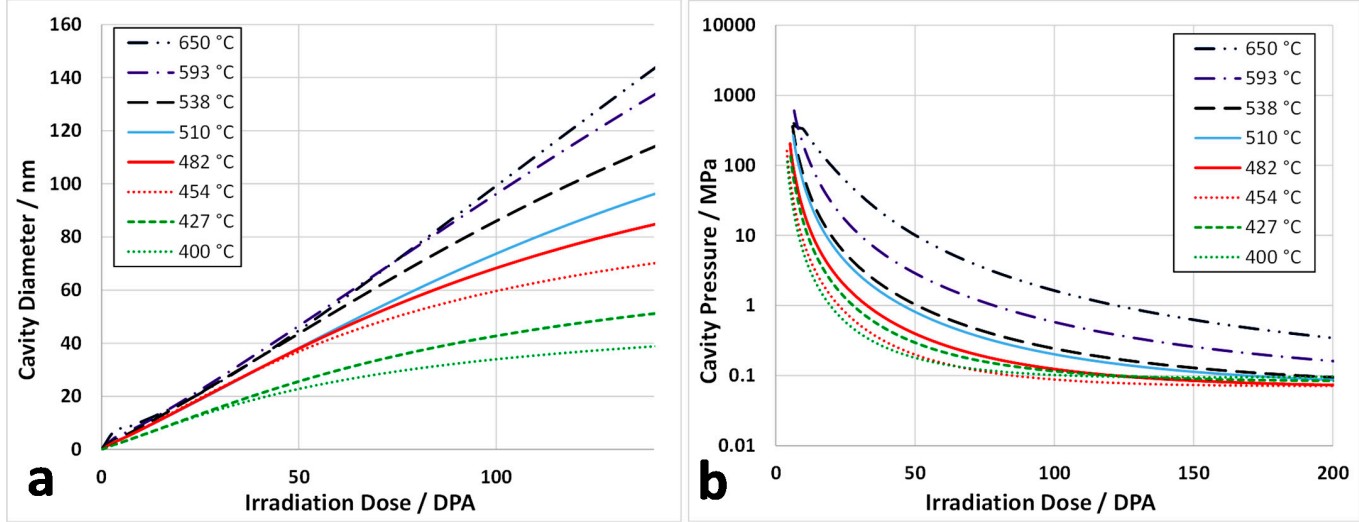

**Figure 17.** Calculated evolution of cavity properties during irradiation in Row 2 of EBR-2: (**a**) Cavity diameters as a function of DPA: (**b**) Cavity pressures as a function of DPA.

The plots in Figure 16 show how the He/V ratio relates to the propensity for emission/absorption of vacancies (the activation energy for vacancy emission) by the evolving cavity size and He content. Figure 17 shows the corresponding average cavity diameters and pressures. For the temperatures in question the He/V and pressures are high early in the irradiation and decrease with increasing dose as the cavities grow by absorption of an excess of vacancy point defects, consistent with what would be expected for bias-driven cavity growth. The initial high pressure arises because of the assumption that the starting condition for the cavities is equivalent to having one He atom per vacancy in the cavities for a given cavity number density at each temperature. The combined effect of increasing cavity diameter, coupled with the pressure due to the He shown in Figure 17, is to affect the stability of the cavities, i.e., the propensity to shrink or grow. It is clear from Figure 16b that all cavities are over-pressurised early on in the irradiation, thus making it more difficult for vacancies to be emitted (high activation energy for vacancy emission). Thereafter the activation energy is negative but has a decreasingly smaller magnitude with increasing

dose for doses >5 DPA. The negative value means that the cavities are under-pressurised and will tend to shrink by emitting more vacancies than are absorbed from the continuum once the irradiating flux is terminated. The growth of the cavities (swelling) shown in Figure 17a occurs because of the non-equilibrium flux of vacancy point defects. The rate of cavity growth slows with increasing dose and this slowing is more prevalent at low temperatures (Figure 17a). The variation in the cavity growth rate (swelling given that the number density is constant) is strongly dependent on the evolution in the cavity sink strength relative to the dislocation density (sink strength) at each temperature. The cavity sink strengths become comparable to the dislocation sink strengths at doses of about 5 DPA at 400 °C, 20 DPA at 500 °C and 60 DPA at 600 °C and, in principle, correspond with the peak swelling rates (see Figure 9b). As the cavity sink strength increases further with increasing dose the cavity growth rate slows.. The cavity diameters shown in Figure 17a conform with the diameters reported by Bhattacharya and Zinkle [1], shown in Figure 8a.

## 8. Conclusions

The dimensional stability of austenitic stainless steels is largely dependent on cavity (void) swelling, which is also dependent to some extent on having He present to stabilise cavities against collapse to vacancy dislocation loops. The swelling that is observed increases with increasing temperature at reactor operating temperatures >250 °C. There is a peak in the measured swelling rate per unit dose at a temperature between 450 °C and 500 °C. A rate theory model based on balance equations has been developed and validated against swelling data from 20% cold-worked 316 SS irradiated in EBR-2. An assessment of the model shows that agreement can be achieved with the selection of the appropriate input parameters for microstructure from various published sources. The best agreement can be obtained with one empirical adjustment involving a positive shift of 0.5 DPA per degree for each degree below 500 °C, which assumes there is an incubation dose for cavity nucleation at low temperatures, <500 °C. The temperature dependence for the swelling is related to the effect of temperature on: (i) dislocation density; (ii) cavity number density; (iii) point defect (vacancy) mobility; and (iv) the internal pressure within the cavities (also dependent on He content), especially at high temperatures (> 500 °C).

In nuclear engineering there are often cases where there is insufficient information to formulate a fully mechanistic model and some degree of empiricism is often applied to model material behaviour in reactor. In this paper we have described a model for cavity swelling that can adequately apply to 20% cold-worked 316 SS irradiated in row 2 or EBR-2. The model so derived can now be applied to other materials and irradiation conditions.

**Supplementary Materials:** The following are available online at https://www.mdpi.com/article/10.3390/jne2040034/s1, Figure S1: Uniaxial tensile yield stress data used for calculating swelling for austenitic stainless steels (316 SS and 304 SS) irradiated and tested at 330 °C. Modified from [51]. The yield stress is shown on the right hand ordinate and the relative dislocation density assuming that the change in yield stress is linearly related to the change in dislocation density, which saturates at a maximum value at a given dose, on the left hand ordinate. The model is not intended to be mechanistic in any way but can serve as a means of including an evolution term for the dislocation density represented by the yield stress. Figure S2: Calculated variation in interstitial bias for dislocations in 316 SS as a function of irradiation temperature during irradiation in EBR-2. Figure S3: Swelling determined by density change as a function of irradiation temperature and dose, as observed in 20% cold-worked AISI 316 irradiated in the EBR-2 fast reactor. All measurements at a given temperature were made on the same specimen after multiple exposures with subsequent reinsertion into the reactor. This procedure minimized specimen-to-specimen data scatter and assisted in a clear visualization of the post-transient swelling rate. Reproduced (with permission) from [2]. Copyright Elsevier, 2020. Figure S4: Measured and predicted volume swelling rates for cold-worked 316 stainless steel irradiated in EBR-2, row 2 at various temperatures and doses for a cavity number density corresponding to the upper dashed line in Figure 8, a cavity surface energy = 2 J·m$^{-2}$ and two different models for FMD production (see text). Figure S5: Measured and predicted volume swelling rates for cold-worked 316 stainless steel irradiated in EBR-2, row 2 at various temperatures

and doses for two variations in cavity number density (rho1) and two different models for FMD production (see text): (a) cavity surface energy = 1 J·m$^{-2}$; (b) cavity surface energy = 2 J·m$^{-2}$.

**Author Contributions:** Conceptualization, M.G.; methodology, M.G., J.R.-N., L.G.; software, M.G., L.G.; validation, M.G., J.R.-N., L.G.; formal analysis, M.G.; investigation, M.G., J.R.-N., L.G.; resources, M.G.; data curation, M.G.; writing—original draft preparation, M.G.; writing—review and editing, M.G., J.R.-N., L.G.; visualization, M.G.; supervision, M.G.; project administration, M.G.; funding acquisition, M.G. All authors have read and agreed to the published version of the manuscript.

**Funding:** This research received no external funding.

**Institutional Review Board Statement:** Not applicable.

**Informed Consent Statement:** Not applicable.

**Data Availability Statement:** Not applicable.

**Acknowledgments:** The authors are grateful to Frank Garner, Steven Xu and Robert Alicandri for useful discussions.

**Conflicts of Interest:** The authors declare no conflict of interest.

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
