# Peer review of "A Rate Theory Model of Radiation-Induced Swelling in an Austenitic Stainless Steel"

_jne, doi:10.3390/jne2040034_

Round 1

Reviewer 1 Report

The manuscript applies a rate theory model to describe swelling behavior in 316 SS metal. Overall, the paper is well written and highly detailed in an attempt to walk the reader through the development of the model and treatment of different parameters and assumptions.  I really like how the paper takes a very systematic approach to recapping prior work, discussing the influence of each mechanism, and then presents the sensitive analysis of the model to outline how they influence the outcome.

The following are items that the authors should take under recommendation to improve the impact of the paper and make it more understandable for the reader:

  • I would like to better understand the “bigger picture” objective of the manuscript. Sure, you are “...describ(ing) a conventional rate-theory model applied to void swelling of 316 stainless steel…”, but what is the big capability gap that is being closed by this work. Are you training the model with the existing data set to enable the model to be predictive for future work? Is the methodology presented here likely transferrable to other alloys or other irradiation conditions?
  • Organization of the Methods and Analysis section 2.1 is a little unconventional. It would be helpful to introduce an additional first section clearly highlighting the “Material and Irradiations” that are specifically analyzed in this study. In this section, you can include Table 1 and info from the last paragraph of the Introduction section. Furthermore, the manuscript occasionally branches out to discuss Ni-alloys and Zr-alloys in various places. It is not always clear why these comparisons are relevant (Fig. 3?), and it may not be clear to the reader that the modeling is focused only on 316 SS.
  • Since the paper is so long and detailed, the reader can get bogged down in the details without a clear vision about where the paper is going. As a result, a paragraph at the end of the introduction which briefly explains the approach taken and foreshadows what the paper will deliver will be helpful for the reader to prepare for what is coming next.
  • The authors might consider utilizing a “Supplementary Materials” to help capture the details, while keeping the manuscript focused on the important items. At times, the paper feels like it is rehashing commonly known themes or getting lost in minutiae. A more concise manuscript may improve reader comprehension and retention. It is up the authors to decide how to manage this.
  • The formatting of the Figures is very poor and inconsistent. Primarily, each figure is formatted differently than the last and the use of labels and colors needs work. At a minimum, the authors need to resolve this prior to publication with a comprehensive strategy for all figures. Examples:
  1. Figure 3 and Figure 4 use different formats. The axis labels are different colors (black vs. gray).
  2. Figure 5 is smaller than Fig. 3 and Fig. 4. Is this by intent? It is recommended to create consistency in sizes between similar figures. Figure 6 is much bigger. Figures 8 and 9 are probably bigger than necessary.
  3. Figure 7 labeling on the horizontal axis is on the plot. Recommend moving this to the bottom of the plot.
  4. The vertical axis label on Figure 9 is potentially incorrect. Should it be “interstitial bias” or similar?
  5. Figure 10 is very difficult to read.
  6. Figure 12a – The symbol for the Cavities in the legend should be a blue circle instead of a black circle to be consistent with the image.
  7. Figure 12b – This image is difficult to interpret and draw the same conclusion for the maximum swelling rate. Since this is an important concept, a better image is merited here.
  8. Figure 13 – This looks like a powerpoint slide. Recommendation is to visualize this in a form of a flow chart with boxes around each step (or similar) and add a border or other differentiating feature.
  9. Figure 14 – Again, dissimilar formatting/sizing from prior plots.
  10. Etc…
  • The description of Fig. 7 is slightly misleading in the text, which states, “Extracting the yield stress data one can derive a fit to the maximum yield stresses as a function of temperature”. In reality, some of yield stresses decrease with irradiation at temperatures above 500 C. The text would be more accurate to refer to the “steady-state yield stress upon irradiation” or similar.

Author Response

Thank you for taking the time to review the manuscript.  The comments and advice were excellent and we appreciate the opportunity to improve the manuscript accordingly.  A revised manuscript has been uploaded together with our response to the detailed review comments.

Reviewer 2 Report

General review comments: This manuscript is well-written but too lengthy as a research paper. However, it needs revision before publication. The following reviews and comments need to be addressed.

Reviews and comments: 

  • In the abstract and even in the introduction, the research gaps in rate theory models are mentioned. However, it is not mentioned which gaps are covered in this paper.
  • Need approximations and simplifications are considered for the used model(s).
  • There were unnecessary gaps between two sentences in the same line. A “.” before the conclusion section heading.
  • Please provide an overview of the paper in the last part of the introduction.
  • Some figures are taken from other sources with scan/snapshot. Please recreate the figures or cite the sources.
  • There are some figures common in the author’s previous publications. These figures should be cited accordingly. Please specify why it is important to use the figures in this manuscript instead of citing.
  • Figure 13 need revision. It should be presented in a flow/block chart format.
  • The formatting of the equations is not appropriate. It might get broken in latex-word conversion. It is recommended to keep the size of the figures, legend, and caption consistent.
  • It is mentioned that there are many rate theory models that were developed. Are there similar models developed by the previous researchers? Presenting the findings/improvement of the used models compared to previous models in quantitative relations would support the readers.
  • What are the accuracy and uncertainty of the result(s) in the listed study? The test data should be present with error bars.
  • The conclusion part should not include any references. Instead, please focus on the contribution of the studies, major findings, challenges, and future research scope.

Author Response

Thank you for taking the time to review the manuscript.  We appreciate the opportunity to improve the manuscript as you have suggested.  A revised manuscript has been uploaded together with our response to the detailed review comments.
